# Quantitative single-protein imaging reveals molecular complex formation of integrin, talin, and kindlin during cell adhesion

Lisa S. Fischer [1,2,7], Christoph Klingner[1,2,7], Thomas Schlichthaerle[3,4,7], Maximilian T. Strauss [3,4,5], Ralph Böttcher [6], Reinhard Fässler [6✉], Ralf Jungmann [3,4✉] & Carsten Grashoff [1,2✉]

Single-molecule localization microscopy (SMLM) enabling the investigation of individual proteins on molecular scales has revolutionized how biological processes are analysed in cells. However, a major limitation of imaging techniques reaching single-protein resolution is the incomplete and often unknown labeling and detection efficiency of the utilized molecular probes. As a result, fundamental processes such as complex formation of distinct molecular species cannot be reliably quantified. Here, we establish a super-resolution microscopy framework, called quantitative single-molecule colocalization analysis (qSMCL), which permits the identification of absolute molecular quantities and thus the investigation of molecular-scale processes inside cells. The method combines multiplexed single-protein resolution imaging, automated cluster detection, in silico data simulation procedures, and widely applicable experimental controls to determine absolute fractions and spatial coordinates of interacting species on a true molecular level, even in highly crowded subcellular structures. The first application of this framework allowed the identification of a long-sought ternary adhesion complex—consisting of talin, kindlin and active β1-integrin—that specifically forms in cell-matrix adhesion sites. Together, the experiments demonstrate that qSMCL allows an absolute quantification of multiplexed SMLM data and thus should be useful for investigating molecular mechanisms underlying numerous processes in cells.

[1] Department of Quantitative Cell Biology, Institute of Molecular Cell Biology, University of Münster, Münster, Germany. [2] Group of Molecular Mechanotransduction, Max Planck Institute of Biochemistry, Martinsried, Germany. [3] Faculty of Physics and Center for Nanoscience, LMU Munich, Munich, Germany. [4] Research Group Molecular Imaging and Bionanotechnology, Max Planck Institute of Biochemistry, Martinsried, Germany. [5] Department of Proteomics and Signal Transduction, Max Planck Institute of Biochemistry, Martinsried, Germany. [6] Department of Molecular Medicine, Max Planck Institute of Biochemistry, Martinsried, Germany. [7] These authors contributed equally: Lisa S. Fischer, Christoph Klingner, Thomas Schlichthaerle. ✉email: faessler@biochem.mpg.de; jungmann@biochem.mpg.de; grashoff@wwu.de

The continuing development of super-resolution (SR) microscopy methods has transformed how cell biological processes are analyzed in cells. Especially techniques with the capability to resolve individual proteins with a resolution in the single-digit nanometer (nm) range, such as DNA-PAINT[1,2] and MINFLUX[3], promise the analysis of subcellular processes with unprecedented molecular detail. However, a major limitation of currently available single-molecule localization microscopy (SMLM) techniques is that only a fraction of all target molecules are imaged. Recent studies estimated that owing to imperfect labeling (LE) and detection efficiencies (DE), only 25–75% of molecules are observed[4,5]. Thus, despite the exquisite spatial resolution that current SMLM techniques offer, it remains a challenge to use these methods for developing an absolute quantitative understanding of intracellular processes. Protein–protein interactions and the formation of macromolecular complexes, for instance, are typically still analyzed with diffraction-limited methods such as proximity ligation assays[6,7], fluorescence complementation procedures[8], or Förster resonance energy transfer-based approaches[9].

The relevance of this shortcoming is especially obvious in focal adhesions (FAs), densely packed, macromolecular structures that connect the extracellular matrix (ECM) with the intracellular actin cytoskeleton[10,11]. FAs are thought to form after the engagement of integrin receptors with two intracellular molecules[12,13], called talin and kindlin, and extensive biochemical and genetic analyses indicate that the interplay of these three molecules is essential for cell adhesion and FA formation[14,15]. Despite their central role in development and survival of metazoans[14,16], however, it is still unclear how individual integrin, talin, and kindlin molecules assemble in cells. A previous SR landmark study used interferometric photo-activation localization microscopy (iPALM) to reveal a horizontal layering of FA proteins[17]; yet, these experiments lacked true molecular resolution and thus neither the overall densities of FA molecules nor their lateral nanoscale arrangement at the plasma membrane is known. Another study applying single-protein tracking PALM investigated the mobility of integrin and talin molecules in cells[18], but if, where, and when distinct integrin receptors engage with talin and/or kindlin, and whether both activators can engage the cytoplasmic integrin tail simultaneously remained unclear.

To allow the application of SMLM to such questions, we here establish a quantitative single-molecule colocalization (qSMCL) analysis that gradually builds upon quantitative single-protein imaging, cluster detection, widely applicable control probes, and theoretical simulations to determine absolute molecular quantities (Fig. 1a). When applied to talin, kindlin, and integrin, qSMCL reveals the molecular densities and spatial arrangement of these molecules, it allows a molecular-scale evaluation of their local complex formation upon cell adhesion, and quantitative analysis of three-target SR imaging data confirms the presence of a long-sought ternary integrin-talin-kindlin complex in FAs.

## Results

### The combination of quantitative PAINT (qPAINT) analysis and control probes allows the quantification of target molecules in crowded subcellular environments

Our framework is based on a combination of genetically encoded molecular probes and DNA-PAINT imaging[1,5], which uses the sequence-specific and transient binding of fluorescently labeled DNA oligonucleotides to their complementary "docking" strands attached to the protein of interest. These short-lived repetitive binding events create the "blinking" that can be harnessed for SMLM. Over the course of image acquisition, these cumulative binding events form a localization cloud, as illustrated in Supplementary Fig. 1a.

As proof of concept, we decided to unravel the lateral molecular organization of talin-1 in FAs. To achieve this, we genetically inserted a HaloTag into a previously validated talin-1 insertion site after the integrin-binding FERM domain at amino acid 447 (talin-Halo447; Supplementary Fig. 2a) and stably expressed the construct in cells[19] genetically depleted of talin-1 and talin-2. We then seeded these fibroblasts onto fibronectin (FN)-coated glass slides and targeted the HaloTag using a chloroalkane (CA)-modified DNA-PAINT docking strand. Subsequent image acquisition in the presence of the complementary, Cy3b-labeled imager strand revealed distinct talin-1 localization clouds in FAs and the free membrane area (MEM; Fig. 1b). This procedure allowed us to distinguish localization clouds as close as 10–15 nm with an overall localization precision[20] of ~7 nm (Fig. 1c, d and Supplementary Fig. 2c).

A major challenge for current SR strategies approaching single-protein resolution, including DNA-PAINT or MINFLUX, is to precisely determine how many copies of a protein reside within a detected localization cloud. To quantify the exact number of talin-1 molecules per localization cloud, we implemented qPAINT analyses[21] by placing DNA origami nanostructures with a defined number of docking strands next to talin-Halo447-expressing cells (Fig. 1e–g). After image acquisition, we used the observed binding kinetics to estimate the imager strand influx rate on single binding sites associated with DNA origami, and utilized this calibration to determine the number of binding sites in talin localization clouds[21] (Supplementary Fig. 1b). This analysis revealed similar values for the single binding sites on DNA origami and talin-1 localization clouds (DNA origami: $1 \pm 0.3$ binding sites; talin-1: $1 \pm 0.4$ binding sites; Fig. 1h, i), indicating that individual talin-1 molecules are detected (Supplementary Fig. 3). As control measurements, we imaged DNA origami with one and three docking sites, which yielded the expected increase in binding site quantity (Fig. 1j–m).

To confirm these observations with an internal probe that also localizes within FAs, we generated a genetically encoded calibration control (CalC), in which a SNAP and a HaloTag are separated by seven amino acids. We inserted this cassette into talin-1 after amino acid 447 (talin-CalC), expressed the constructs in talin-deficient cells, labeled both tags with the same docking strand, and imaged with DNA-PAINT. As expected, quantification of single- and dual-labeled talin-CalC localization clouds yielded either one or two binding sites (single-labeled: $1.1 \pm 0.3$ binding sites; dual-labeled: $2.1 \pm 0.6$ binding sites; Fig. 1n, o). Together, these measurements demonstrate that the combination of DNA-PAINT, qPAINT, and CalCs allows the identification of single proteins in highly crowded cellular environments. In this case, the experiments demonstrate that the observed localization clouds of talin-Halo447 represent individual talin-1 molecules.

### Cluster analysis and theoretical simulations can unravel the molecular organization of target proteins

The inability to decipher how distinct adhesion proteins are laterally organized on the molecular scale at the plasma membrane has been a key obstacle towards understanding FA function. A major problem in obtaining such information from methodologies with single-protein resolution is to extract quantitative information from single-molecule localization data[22,23]. To overcome this challenge, we next developed an automated data processing procedure that was based on a DBSCAN (density-based spatial clustering of application with noise) cluster detection[24] using constant parameter sets enabled by highly reproducible data acquisition (Supplementary Fig. 4). The algorithm includes two consecutive filtering steps to remove all unspecific signals and calculate the nearest-neighbor distance (NND) between individual proteins

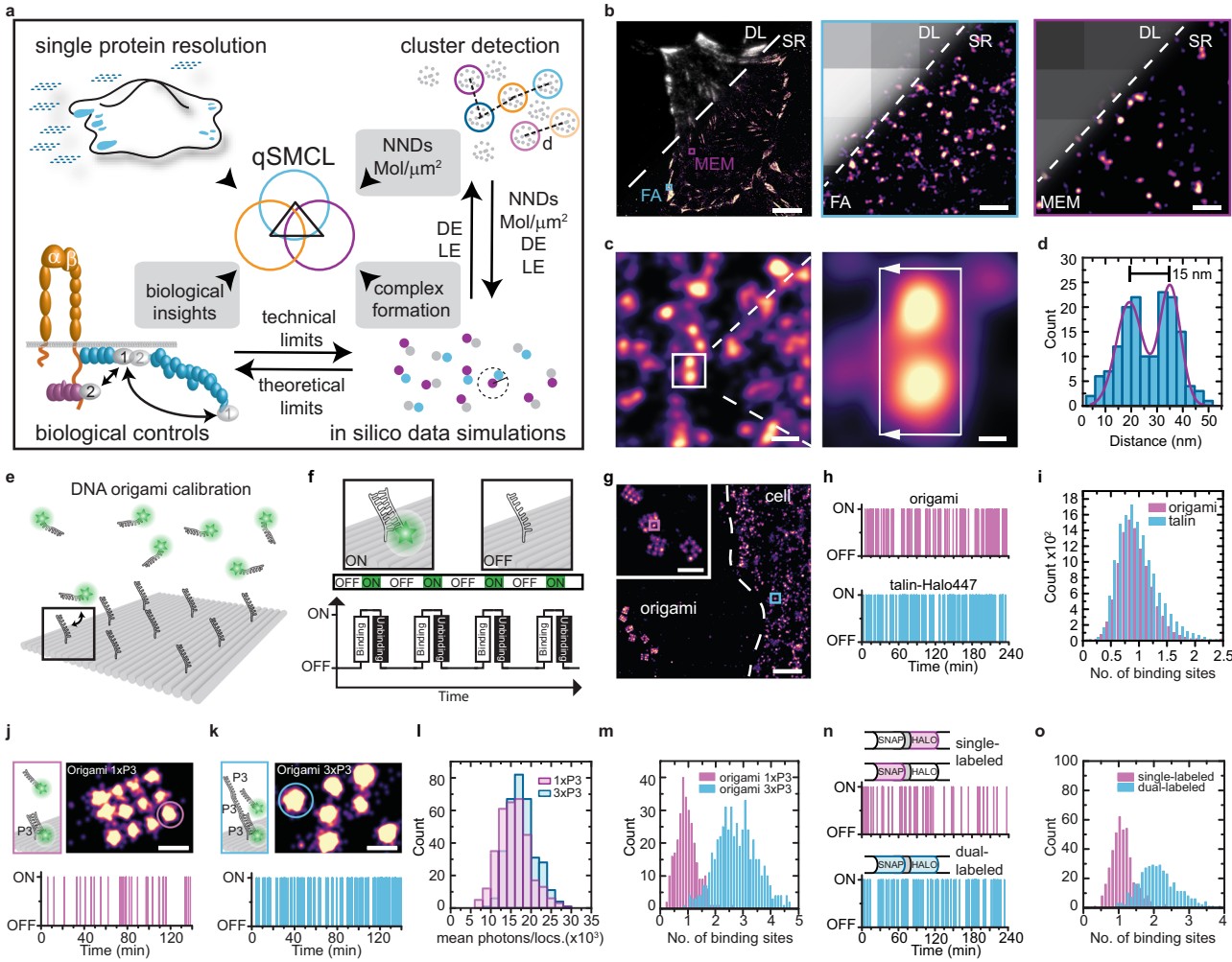

**Fig. 1 Resolving the location of individual molecules in crowded subcellular structures. a** Schematic overview of quantitative single-molecule colocalization analysis (qSMCL) to evaluate nearest-neighbor distance (NND), molecular densities (mol/μm²), detection efficiencies (DEs), and labeling efficiencies (LEs). **b** Overlay of diffraction-limited (DL) and super-resolved (SR) DNA-PAINT image showing a talin-Halo447 expressing cell. Zoom into focal adhesion area reveals distinct talin localization clouds. Regions in the free membrane (MEM) are characterized by more disperse talin localization clouds. **c** The approach allows the separation of distinct talin-1 localization clouds at a distance of approximately 15 nm. **d** Histogram analysis demonstrates that the localization clouds shown in **c** can indeed be resolved ($\sigma_{Peak1} = 3.5$ nm, $\sigma_{Peak2} = 5.6$ nm); number of localizations ($n_{locs} = 157$). **e** Schematic illustration of a DNA origami calibration. **f** Transient binding events of dye-labeled imager strands to the complementary, immobilized docking strands, creating the characteristic "blinking" (ON/OFF) required for single-molecule localization microscopy. **g** DNA origami structures carrying single docking strands were placed next to talin-Halo447 cells to calculate the number of molecules per localization cloud. Inset: Zoom onto DNA origami structures. **h** Plotting the binding events over the number of recorded frames reveals similar binding traces in DNA origami and talin-Halo447 localization clouds. **i** Quantitative histogram analysis of absolute binding site numbers on DNA origami and talin localization clouds confirming that the observed talin localization clouds represent single talin-1 proteins ($n = 8$ cells). **j** Zoom-in of a 20 nm grid DNA origami displaying a single binding sequence (1xP3) per site and the corresponding binding event history. **k** Zoom-in of a DNA origami structure with three concatenated binding sequences (3xP3) per docking strand and the corresponding binding event history. **l** Analysis of the mean photon counts per individual localization event reveals highly similar values for single (**j**) and triple (**k**) binding sequences ($n = 334$ localization events). **m** qPAINT analysis confirmed either one or three binding sites per DNA origami localization cloud ($n = 1$). **n** Binding frequency of a single-labeled (due to <100% labeling efficiency) and a dual-labeled localization cloud. **o** qPAINT analysis reveals, as expected, that either one or two binding sites are detected ($n = 1$ cell). Scale bars: 7 μm (**b**), 370 nm (**g**), 110 nm (**g** inset), 70 nm (**b** insets), 50 nm (**c**, right), 30 nm (**j**, **k**), 9 nm (**c**, right). Source data are provided in the Source Data file.

(Supplementary Fig. 5). This procedure was complemented with theoretical simulations to determine true molecular quantities. To validate the data processing algorithm, we performed control measurements using a set of DNA origami, in which binding sites were separated by 20–35 nm. These experiments confirmed that distances down to 25 nm can be reliably separated with this postprocessing data analysis (Supplementary Fig. 6), and we applied the procedure to evaluate how talin-1 molecules assemble at the plasma membrane during the initiation and maturation of cell–matrix adhesions.

At the initial phase of cell attachment, and in the absence of identifiable adhesion complexes, single talin-1 molecules appeared evenly spaced at the plasma membrane with an average NND of ~95 nm. After 15–25 min, when talin-1 started to organize in small adhesion sites, these distances reduced to ~55 nm and further condensed with the onset of anisotropic cell spreading to ~45 nm (Fig. 2a, b). This NND value then remained constant and was consistently observed in FAs of fully spread cells 16 h after seeding, even though cells adopt heterogeneous morphologies under these conditions. Intriguingly, a fraction of

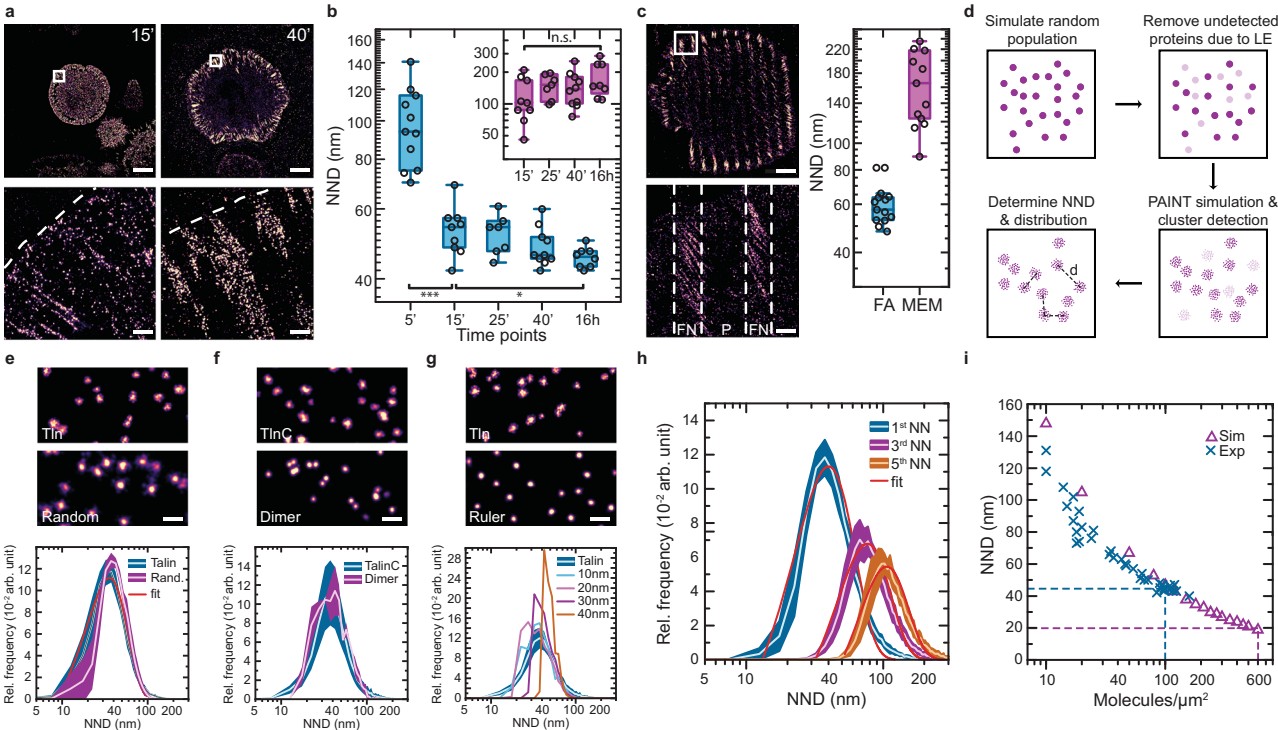

**Fig. 2 Combining cluster analysis with theoretical simulations enables the investigation of molecular assembly models. a** Localization of talin-1 to first adhesion clusters 15 min and maturing focal adhesions (FAs) 40 min after initiation of cell–ECM adhesion. **b** Nearest-neighbor distance (NND) analysis reveals the compaction of talin-1 molecules during FA maturation towards a characteristic endpoint density (blue); the molecular distance of talin-1 in the membrane region (MEM) is unaffected by the cell adhesion state (purple) ($n_{5\,min} = 11$; $n_{15\,min} = 9$; $n_{25\,min} = 7$; $n_{40\,min} = 10$; $n_{16\,h} = 8$ cells) (FA: $p_{5\,min\ vs.\ 15\,min} = 1.79 \times 10^{-5}$; $p_{15\,min\ vs.\ 16\,h} = 0.01619$; MEM: $p_{5\,min\ vs.\ 16\,h} = 0.06115$). **c** Analysis of talin-Halo447 expressing cells on 1 μm thick micropatterned fibronectin (FN) stripes—separated by passivated (P) 2 μm stripes—demonstrates that the molecular localization of talin is governed by integrin-mediated ECM engagement ($n = 13$ cells). **d** Schematic overview of the DNA-PAINT data simulation workflow considering random distributions, labeling efficiency (LE) and cluster detection efficiency (DE) allowing distance ($d$) calculations. **e** Comparison of experimental talin-Halo447 data (Talin) with a simulation of randomly organized proteins (Rand.) shows highly similar distributions. Bottom: experimental data of talin-1 were fitted with a 2D Poisson density function (red line) and plotted as relative frequency (Rel. frequency) in arbitrary units (arb. unit). **f** Comparing experimental data from C-terminally tagged talin-1 constructs (TalinC) with a simulation of a protein complex (Dimer) does not indicate talin dimer formation. **g** Comparison of experimental talin-Halo447 data (Talin) with a simulation assuming steric hindrance at 10–40 nm indicates that talin does not act as a molecular ruler in FAs. **h** Talin-1 distribution of the 1st, 3rd, and 5th nearest-neighbor (NN) in FAs. Distributions were fitted with a 2D Poisson density function (red line) indicating a random organization of talin-1 molecules on length scales between 40 and 120 nm ($n = 8$ cells). **i** Random distribution simulations (Sim; magenta triangle) indicate an absolute molecular density of approximately 600 molecules/μm² for talin-1 in focal adhesions (purple dashed line); blue crosses indicate experimental data (Exp) and the dashed blue line indicates the experimentally observed densities ($n = 39$ cells). Boxplots show median and 25th and 75th percentage with whiskers reaching to the last data point within 1.5× interquartile range. NND distributions show the mean (line) ± SD (shaded area). Two-sample $t$-test: ***$p \leq 0.001$, *$p \leq 0.05$, n.s. (not significant) $p > 0.05$. NND distributions show mean and standard deviations. Scale bars: 10 μm (**a**, **c**), 500 nm (**a**, **c**, inset), and 50 nm (**e–g**). Source data are provided in the Source Data file.

talin-1 was consistently found outside FAs in the MEM at distances >100 nm, a value that appeared largely insensitive to the cell adhesion state (Fig. 2b). Consistent with our experiments above, qPAINT measurements indicated individual proteins per localization cloud at all time points (Supplementary Fig. 7).

To confirm that the results were independent of the employed labeling strategy, we used talin-deficient cells reconstituted with talin-YPet (talin-YPet447)[25] and labeled them with a GFP-nanobody (GFP-NB) that was conjugated to a DNA-PAINT docking strand[26]. Subsequent DNA-PAINT imaging revealed NNDs in FAs and the MEM that were indistinguishable from talin-Halo447 data sets (Supplementary Fig. 8a–c). As a final test, we confirmed that talin-1 molecules assemble in an integrin receptor-dependent manner. We seeded talin-Halo447 cells onto micropatterned surfaces featuring FN-coated stripes, which were interspaced with passivated areas that integrin receptors cannot engage. As expected, talin-1 assembled at short molecular distances in FAs on FN-stripes and at large distances in passivated areas (Fig. 2c). Together, the data demonstrate that

cell adhesion formation is characterized by the gradual, and integrin receptor-dependent condensation of individual talin-1 proteins culminating in a characteristic molecular density.

To understand the underlying spatial distribution of talin-1 molecules in more detail, we evaluated different assembly models that match the observed talin-1 NND distribution by simulating data assuming different patterns for the organization of talin-1 molecules and considering LE and DE. According to a previous study[4], the LE of HaloTag was set to 30%, and the DE for the DBSCAN-based analysis was estimated, based on a comparison with manually selected data, to be 50–60% (Fig. 2d). Intriguingly, the data were best described by a homogeneous Poisson point process, indicating—at least down to our detection limit—a random organization of molecules in FAs (Fig. 2e–g). In addition, we analyzed the data for the presence of higher-order assembly patterns or clusters by evaluating distances to the third and fifth talin-1 protein, covering a spatial range of 40–120 nm. These data sets were also consistent with a random organization of molecules indicating that talin-1 does not assemble in lateral FA

substructures under these conditions (Fig. 2h). Finally, we estimated the absolute molecular density of talin-1 in FAs of cells 16 h after seeding (Fig. 2i). We first mimicked lower molecular densities by successively reducing the docking strand concentration in our experiments. We then confirmed that the resulting data sets are described by a simulation using the experimentally obtained NNDs and the above-mentioned estimates of LE and DE. We then extrapolated the data by simulating a gradual increase of LE and DE. These simulations predicted an absolute talin-to-talin distance of 20–25 nm in mature FAs (16 h) with a molecular density of ~600 talin-1 molecules/μm$^2$ (Fig. 2i).

Altogether, this demonstrates that the underlying spatial distribution of individual molecules and their absolute molecular densities can be determined by implementing advanced cluster detection and theoretical simulations. In the case of talin-1, the data demonstrate that the organization of proteins is not set by a previously hypothesized function of talin as a molecular ruler[27], but primarily governed by the molecular density in the adhesion area.

**Integrating Exchange-PAINT with in silico data simulations allows the analysis of protein–protein interactions and talin–kindlin complex formation.** Cell–ECM adhesion requires the engagement of integrin receptors with not only talin but also kindlin, and it has been demonstrated that the FERM domains of talin and kindlin bind the cytoplasmic tail of β-integrins at two adjacent but distinct motifs[14,15]. It is unclear whether both proteins can co-assemble with integrin receptors in cells, yet approaches for applying SMLM techniques to address this issue in a quantitative fashion are missing. Therefore, we generated an N-terminal SNAP-tagged kindlin-2 construct (SNAP-kindlin; Supplementary Fig. 2b), co-expressed it with talin-Halo447 in fibroblasts deficient for kindlin-1, kindlin-2, talin-1, and talin-2, and performed Exchange-PAINT experiments[2] (Supplementary Figs. 8d, e and 9). As expected, kindlin-2 was observed at high densities in FAs. We detected intermolecular distances of ~55 nm for kindlin-2, while talin-1 assembled with an NND of ~40 nm in these cells (Fig. 3a–c). Intriguingly, individual talin-1 and kindlin-2 molecules were frequently observed in close proximity with an average "kindlin-to-talin" distance of <30 nm (Fig. 3c, d). To validate this observation and ensure that results are independent of the labeling approach, we reconstituted kindlin- and talin-deficient cells with Halo-kindlin and talin-SNAP447. Exchange-PAINT yielded highly similar results with largest NNDs for kindlin-2, slightly shorter NNDs for talin-1, and seemingly spatially associated talin-1–kindlin-2 pairs (Supplementary Fig. 10).

To explore the spatial proximity of talin and kindlin further, we combined DNA-PAINT experiments with a simulation approach to predict the absolute degree of molecular association (Fig. 3e–g). Based on the observed kindlin-to-talin distances (Fig. 3c), we here define such association between talin and kindlin by an intermolecular distance of ≤25 nm, and we note that the detection limit of the DBSCAN-based analysis (Supplementary Fig. 6) does not influence the colocalization analysis as both proteins are independent entities. However, the key problem for all available SMLM approaches is that a significant fraction of target molecules, typically in the order of 25–75%, remain undetected[4]. The reason is imperfect labeling and inefficient cluster detection, which severely complicates reconstructing the true molecular architecture of still unexplored subcellular structures, or the degree of spatial association between individual proteins. We thus established a theoretical framework that accounts for the experimentally observed protein distributions, molecular densities, and previously determined LEs for HaloTag (30%) and SNAP-tag (20%) avoiding bias through the single-molecule

localization process[5]. In addition, we performed theoretical calculations to estimate how different molecular densities and LEs would influence the result of those simulations (Supplementary Fig. 11).

As an experimental validation of this theoretical framework, we used talin-CalC cells, where both tags are only separated by seven amino acids, mimicking a synthetically engineered scenario of perfect spatial proximity and therefore defining the upper boundary of detectable colocalization. For this control, we experimentally observed that ~54% of all SNAP and HaloTag-labeled molecules assemble within 25 nm, which was consistent with theoretical predictions (54 ± 0.01%; Fig. 3e, h). As a measure of unspecific, density-based proximity, we simulated two random distributions using experimentally observed molecular densities of talin-Halo447 and SNAP-kindlin, which indicated that ~35% of talin-1 and kindlin-2 molecules are in close proximity, merely due to the high protein density in FAs (Fig. 3e, h). In contrast, experiments in talin-Halo447 and SNAP-kindlin reconstituted cells revealed that 45% of talin-1 and kindlin-2 were in close proximity. To predict the absolute percentage of proteins, which actively engage in complex formation, we simulated protein distributions with varying degrees of spatial association (0–100%), removed the undetected fractions, and calculated the remaining percentage of spatially associated proteins (Fig. 3f, g). These simulations showed that the experimentally observed colocalization translates into 55% of all talin and kindlin molecules being actively engaged in complex formation (Fig. 3h). To further validate the specificity of this observation, we generated a talin-1 construct, in which the HaloTag was located at talin's C terminus, and co-expressed it with SNAP-kindlin. As expected and consistent with the simulations, this C-terminally tagged talin-1 displayed significantly reduced spatial proximity to kindlin-2 when compared to talin-Halo447 (Fig. 3e).

Together, these data show that the integration of Exchange-PAINT with in silico simulations allow the absolute quantitative analysis of protein–protein interactions with nm-scale resolution. In the case of talin-1 and kindlin-2, these experiments reveal that a majority of talin-1 and kindlin-2 molecules spatially associate upon cell adhesion. Since we did not detect significant molecular proximity of talin-1 and kindlin-2 outside FAs (Fig. 3e), we conclude that the complex formation occurs specifically within the adhesion area.

**qSMCL enables a quantitative complex formation analysis and demonstrates that talin and kindlin spatially associate with active integrin receptors.** In view of the proposed models of integrin activation[12,13], the results above imply that talin-1 and kindlin-2 indeed associate in FAs to induce or maintain the active state of integrin receptors. To test this hypothesis directly in cells with nm-scale resolution, we established three-target Exchange-PAINT experiments, visualizing talin-1 and kindlin-2 molecules together with integrin receptors (Fig. 4a). For the detection of active integrins, we conjugated DNA-PAINT docking strands to the 9EG7 antibody, which binds to the β1-integrin in its extended conformation[28]. Consistent with the previous experiments, talin-1 and kindlin-2 distributions were again characterized by an average molecular distance of ~45–55 nm, whereas active β1-integrin receptors were spaced at larger distances of ~85 nm, presumably because the 9EG7 antibody detects only the activated fraction of all β1-integrin molecules. Moreover, cells used in this experiment express integrin αvβ3, and a fraction of talin and kindlin may be associated with this FN receptor. Nonetheless, we frequently observed β1-integrin localizations in close proximity to talin–kindlin complexes with average integrin-to-kindlin (I2K) and integrin-to-talin (I2T) distances of ~35 nm (Fig. 4b–d).

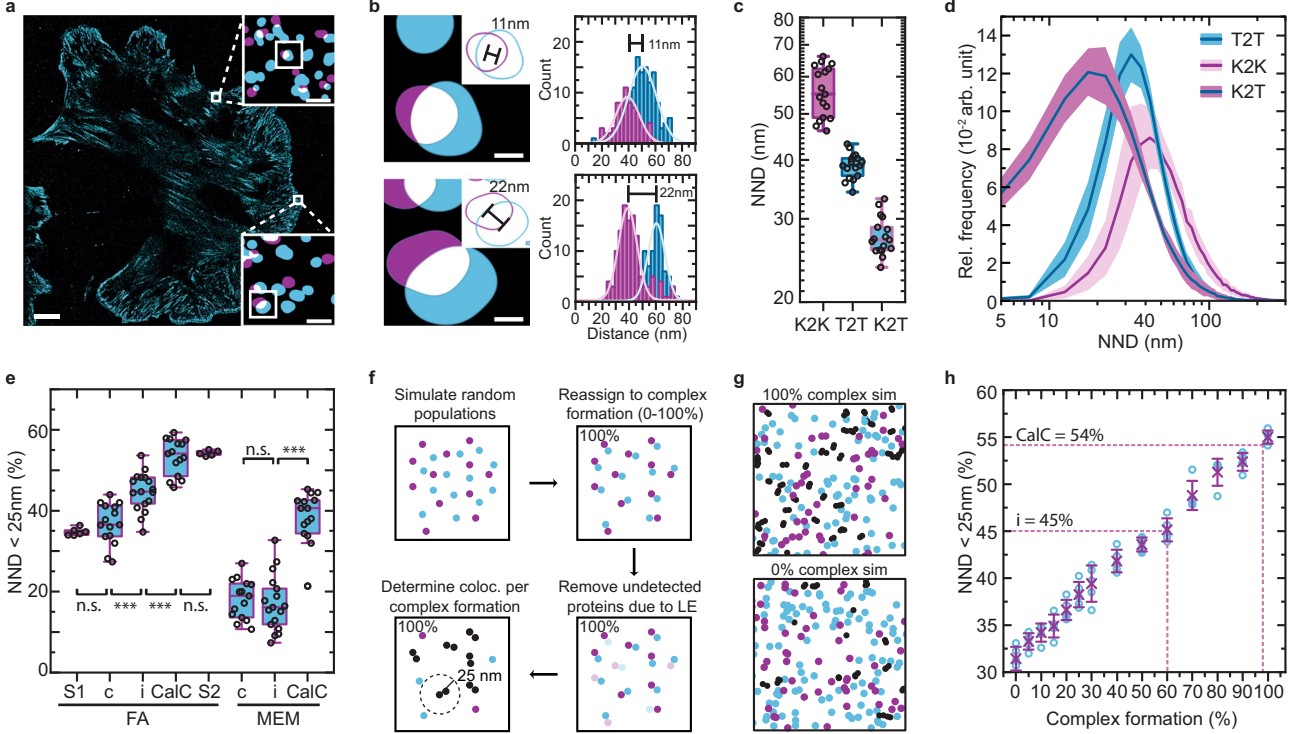

**Fig. 3 Integrating Exchange-PAINT with theoretical simulations allows the quantification of protein–protein interactions. a** Representative image of a reconstituted cell with labeled talin-Halo447 (blue) and SNAP-kindlin (purple). **b** Zoom into focal adhesions reveals that talin-1 and kindlin-2 molecules are in close spatial proximity. Gaussian fits of the aligned single-molecule localizations to their center-of-mass reveal neighboring talin-1 (blue) and kindlin-2 (purple) molecules at distances of 11 nm ($\sigma_{Peak1}$ = 8.8 nm; $\sigma_{Peak2}$ = 10.4 nm) and 22 nm ($\sigma_{Peak1}$ = 6.4 nm; $\sigma_{Peak2}$ = 6.9 nm). **c** Nearest-neighbor distance (NND) analyses reveal the molecular spacing between kindlin-2 (K2K) and talin-1 molecules (T2T). The average distance between kindlin and talin molecules (K2T) is significantly lower ($n = 17$ cells). **d** T2T, K2K, and K2T NND distributions (plotted as relative frequency (Rel. frequency) in arbitrary units (arb. unit)) indicate a shift of K2T towards shorter distances ($n = 17$ cells). **e** Simulations (S1) indicate that 35% of talin-1 and kindlin-2 molecules can be expected in close spatial proximity (<25 nm), due to high protein density in FAs. Forty-five percent of labeled kindlin-2 molecules are in close proximity to the next internally tagged talin-1 molecule (i); this spatial proximity is significantly lower in cells expressing C-terminally tagged talin-1 (c). Experiments of talin-CalC, which mimics perfect spatial proximity, yields a value of 54% (CalC) in consistency with simulations that consider the published labeling efficiencies for HaloTag and SNAP-tag (S2). The observed effects are specific to focal adhesions (FAs) and not observed in free membrane (MEM) regions ($n_{S1} = 6$; $n_c = 15$; $n_i = 17$; $n_{CalC} = 15$; $n_{S2} = 5$ simulated data sets) (FA: $p_{S1\ vs.\ c} = 0.365$; $p_{c\ vs.\ i} = 8.05 \times 10^{-5}$; $p_{i\ vs.\ CalC} = 9.84 \times 10^{-6}$; $p_{CalC\ vs.\ S2} = 0.667$; MEM: $p_{c\ vs.\ i} = 0.61$; $p_{i\ vs.\ CalC} = 4.33 \times 10^{-10}$). NND distributions show the mean (line) ± SD (shaded area). **f** Schematic overview of in silico data simulation for two protein populations (purple and blue) undergoing molecular complex formation, considering the labeling efficiencies (LEs). Spatially associated proteins within 25 nm are colored in black. **g** Simulation data (sim) of two protein populations undergoing 100% or 0% complex formation. **h** Theoretical simulations of colocalization experiments. The percentage of binding sites closer than 25 nm was plotted over the percentage of complex formation. Crosses indicate mean, and error bars the standard deviation. Boxplots show median and 25th and 75th percentage with whiskers reaching the last data point within 1.5× interquartile range. Two-sample $t$-test: ***$p \leq 0.001$, n.s. (not significant) $p > 0.05$. Visualization of **a** and **b** is based on the convex hull of grouped localization clouds. Scale bars: 5 μm (**a**), 100 nm (**a**, insets), and 20 nm (**b**). Source data are provided in the Source Data file.

To corroborate that this observation reflects direct spatial association, we implemented a data analysis routine enabling the analysis of ternary complex formation. We analyzed our experimental data by localizing an active integrin receptor and then calculating its related nearest-kindlin and nearest-talin protein (I2KT). These I2KT tuples were compared with simulated data, in which β1-integrin, kindlin-2, and talin-1 were randomly distributed with the experimentally observed molecular densities. Generating a two-dimensional (2D) heat map of both data sets revealed an enrichment of short-distance I2KT tuples in the experimental dataset when compared to a random simulation of tuples (Fig. 4e) and subsequent bootstrap analysis confirmed that these differences are highly significant (Fig. 4f, g). This final example illustrates that the combination of quantitative single-protein imaging, data simulations and statistical analysis can be utilized to provide direct evidence for the complex formation of multiple molecular species with exquisite spatial resolution in crowded subcellular structures. Here, the analysis shows that β1-

integrin, talin-1, and kindlin-2 undergo a specific spatial association during cell–ECM adhesion.

## Discussion
We here developed an approach that overcomes a number of obstacles common to virtually all SMLM approaches. Our framework creates a path to combine single-protein resolution imaging with automated cluster detection and it shows how widely applicable CalCs and theoretical considerations, including the often ignored undetected protein fraction, allow the quantification of protein–protein interactions and molecular complex formation between multiple molecular species in cells. The approach should be adaptable to all SMLM techniques obtaining true single-protein resolution such as MINFLUX[3], Expansion SMLM[29], and—as shown in this study—PAINT-based approaches. The experiments require a defined number of docking strands per target protein to determine the number of molecules per localization cloud using qPAINT calibration or, potentially,

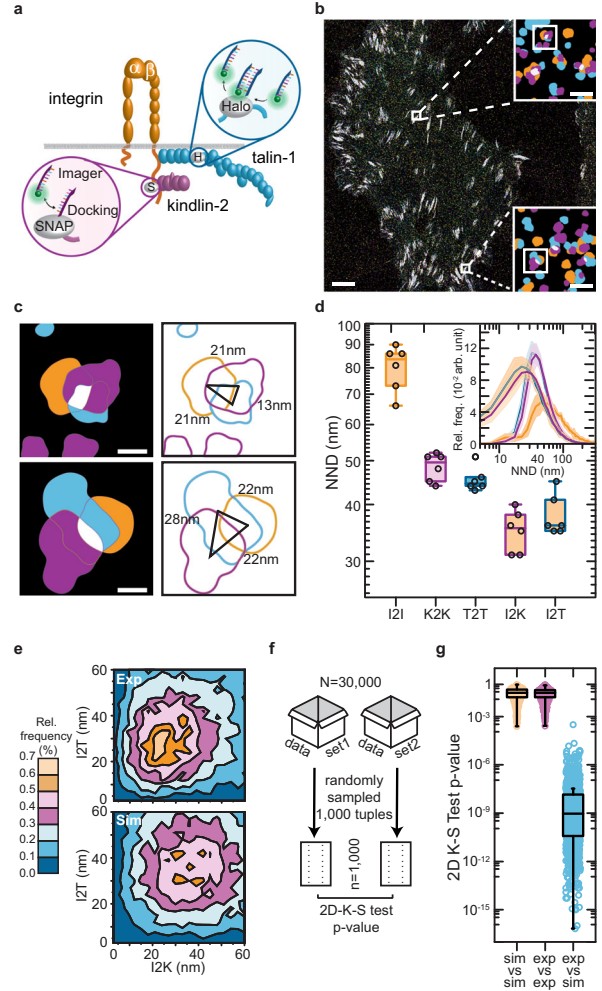

**Fig. 4 qSMCL provides direct evidence for a ternary complex formation of talin and kindlin with active integrin receptors. a** Overview of HaloTag and SNAP-tag-based DNA-PAINT imaging of talin and kindlin molecules in cells. **b** Representative image of labeled talin-Halo447, SNAP-kindlin, and β1-integrin; integrins were labeled with a DNA-conjugated antibody (9EG7) allowing the visualization (but not absolute quantification, due to heterogenous docking site number per labeled antibody) of β1-integrin in the extended conformation. **c** Zoom into FAs reveals close proximity of talin-1 (blue), kindlin-2 (purple), and extended β1-integrin (orange). **d** Nearest-neighbor distance (NND) analyses reveal the molecular spacing distributions of talin-1 (T2T) and kindlin-2 (K2K) in reconstituted cells; extended β1-integrin is observed at larger distances of ~85 nm (I2I); average integrin-to-kindlin (I2K) and integrin-to-talin (I2T) distances were observed at around 35 nm (n = 6 cells). NND distributions show mean (line) ± SD (shaded area). **e** Randomly distributed simulations at the observed molecular densities for talin-1, kindlin-2, and β1-integrin were compared to experimental data by plotting I2K and corresponding I2T tuples for each detected β1-integrin; Rel. frequency: relative frequency. **f** Comparing experimental integrin-talin-kindlin data with simulations using bootstrap analysis. To determine the goodness of fit and significance level between experimental data and simulated data, a 2D Kolmogorov–Smirnov (K–S) test was used. **g** Statistical evaluation of experimental and simulated integrin-talin-kindlin bootstrapped data (sample size = 1000 data points, test runs = 1000) revealed high p-values for intrinsic data bootstrapping ("sim vs. sim" and "exp vs. exp") but low p-values when comparing experimental with simulated data sets ("exp vs. sim"). Boxplots show median and 25th and 75th percentage with whiskers reaching to the last data point within 1.5× interquartile range. Visualization of **b** and **c** is based on the convex hull of grouped localization clouds. Scale bars: 5 µm (**b**), 100 nm (**b**, insets), and 20 nm (**c**). Source data are provided in the Source Data file.

recently developed localization-based fluorescence correlation spectroscopy[30]. Furthermore, the theoretical simulations require an estimate of the LEs and DEs, and biological controls are key to validate the calculated degree of protein–protein association in the given subcellular context.

By applying qSMCL to integrin, talin, and kindlin, we obtain the first molecularly resolved view of the integrin activation machinery showing that talin and kindlin proteins condense upon cell–ECM adhesion to characteristic molecular densities—which presumably depend on the expressed talin[31] and kindlin isoforms[32] and are likely altered in diseased states[33,34]—that facilitate their joint association with integrins. The detection of talin and kindlin located at active integrin receptors by three-target SMLM supports a model, in which either activating integrin receptors or maintaining their active state is a process that involves the spatial association of all three proteins[12,13].

How other FA core molecules[11] associate with this basic integrin activation unit to regulate cell adhesion will certainly be important to explore, and the here developed framework should greatly facilitate these experiments. Furthermore, qSMCL should be directly applicable to a large variety of cell biological questions where molecular distance and proximity assessment or molecular complex formation are key to uncover new biology.

## Methods

**Labeling probes, reagents, and antibodies**. CA (HaloTag ligand)- and benzyl-guanine (BG; SNAP-tag ligand)-modified docking strands carrying an Atto488 dye at the 3′ end, tetrazine- and azide-modified DNA for antibody and NB coupling, were custom ordered from "Biomers.net." Imager strands with a Cy3b modification at the 3′ end were purchased from Eurofins; for oligonucleotide sequences, see Supplementary Tables 1 and 2. For imaging, the following antioxidant stock solutions were used: 40× PCA (protocatechuic acid) solution (154 mg PCA diluted in 10 ml ddH$_2$O, pH 9.0), 100× Trolox (100 mg of Trolox in 430 µl methanol, 345 µl NaOH (1 M), and 3.2 ml ddH$_2$O), and 100× PCD (protocatechuate 3,4-dioxygenase) solution (9.3 mg of PCD diluted in 13.3 ml of 50% glycerol with 50 mM KCl, 1 mM EDTA, and 100 mM Tris-HCl, pH 8.0). PCD, PCA, and Trolox stocks were stored at −20 °C. In addition, the following antibodies and reagents were used: α-integrin 9EG7 (BD Biosciences, 553715, 1:200), GFP-NB (NanoTag, Clone 1B2, 1:200), paraformaldehyde (Roth, 4980.1), Triton X-100 (Roth, 3051.4), bovine serum albumin (Serva, 11930.03), and dimethylformamide (Thermo Fisher, 20673).

**Plasmid construction**. Talin-1 expression constructs are based on human talin-1 complementary DNA (cDNA) (NM_006289). For internal tagging, a linker encoding for 5′ SalI/3′ NotI restriction sites was generated after the base pair encoding for amino acid 447, and HaloTag (Promega), SNAP-tag (New England Biolabs) or the CalC cassette were inserted by Gibson cloning. The C-terminal talin-1 fusion construct was generated using EcoRI/BamHI restriction sites. The assembled cDNAs were then transferred into a modified pLPCX (pLPCXmod) that drives expression through a cytomegalovirus promoter and the correct sequence of all constructs was confirmed by DNA-sequencing (Eurofins Genomics). Kindlin expression constructs are based on mouse kindlin-2 cDNA (Gene ID: 218952), which was tagged N-terminally with either HaloTag or SNAP-tag via Gibson cloning using HindIII/NotI restriction sites. The construct was cloned into pLPCXmod with a crippled cytomegalovirus promoter to avoid overexpression.

**Cell culture**. Cells were maintained in high glucose Dulbecco's modified Eagle' medium (Thermo Fisher, 31966047) supplemented with 10% fetal bovine serum (Thermo Fisher, 10270106) and 1% penicillin/streptomycin (Sigma, P4333). The talin constructs were stably expressed by retroviral infection in double knockout fibroblasts deficient for talin-1 and talin-2 (Tln1−/−Tln2−/−)[19,25], or co-expressed with kindlin-2 constructs in quadruple knockout fibroblasts deficient for talin-1, talin-2, kindlin-1, and kindlin-2 (Tln1−/−Tln2−/− K1−/−K2−/−)[35]. For imaging, 40,000 cells were seeded on ibidi µ-Dishes (ibidi, 81158) with or without coating of 10 µg/ml FN (Calbiochem, 341631).

**Cell fixation and labeling of HaloTag, SNAP-tag, and YPet**. Cells were fixed with pre-warmed 4% paraformaldehyde (PFA) solution for 10 min, washed 3× with phosphate-buffered solution (PBS), and stained with 1 µM of either CA (HaloTag)- or BG (SNAP-tag)-linked docking strands, as summarized in Supplementary Table 1. Staining was performed in PBS containing 0.2% Triton X-100 overnight. For GFP-NB imaging, cells were fixed with pre-warmed 4% PFA solution for 10 min, washed 3× with PBS, incubated for 90 min in 0.2% Triton X-100/4% bovine serum albumin (BSA) in PBS, and stained with P3-conjugated GFP-NB in

4% BSA solution overnight. Cells were then washed 3× for 5 min in 1× PBS, incubated with a 1:3 dilution of 90 nm gold particles (Cytodiagnostics, G-90-100) as drift markers in 1× PBS for 5 min, washed again 3× 5 min in PBS, and immediately imaged.

**SR microscopy**. Fluorescence imaging was carried out on two microscope systems. The first system was based on an inverted Nikon Eclipse Ti microscope (Nikon Instruments) equipped with the Perfect Focus System, applying an objective-type total internal reflection fluorescence (TIRF) configuration with an oil-immersion objective (Apo SR TIRF ×100, NA 1.49, Oil). TIRF angle was adjusted for highest signal-to-noise ratio prior to imaging. For excitation of Atto488 and Cy3b, a 488 nm laser (Toptica iBeam smart, 200 mW) and a 561 nm laser (Coherent Sapphire, 200 mW) was used, respectively. The lasers were fiber-coupled and, after entering the microscope, the laser beam was passed through cleanup filters (ZET488/10× or ZET561/10, Chroma Technology) and coupled into the microscope objective using a beam splitter (ZT488rdc or ZT561rdc, Chroma Technology). Fluorescence light was spectrally filtered with two emission filters (ET525/50m and ET500lp for 488 nm excitation and ET600/50m and ET575lp for 561 nm laser excitation, Chroma Technology) and imaged on an sCMOS camera (Andor Zyla 4.2) without further magnification resulting—after 2 × 2 binning—in an effective pixel size of 130 nm per pixel.

The second system was a commercial Nikon Ti-E N-SIM/N-STORM setup equipped with the Perfect Focus System. The objective was a CFI SR APO TIRF 100xH oil, NA 1.49, WD 0.12 mm (Nikon) using immersion oil from Nikon (nd: 1.515). The TIRF angle was adjusted for highest signal-to-noise ratio prior to imaging. The light source was controlled by an LU-NV Laser Unit with a 488 nm (max. 70 mW at the sample) and a 561 nm (max. 70 mW at the sample) laser line using independent TIRF filter cubes (Chroma filter cube Nikon TIRF 488: 525/50 and Nikon TIRF 561: 605/50). Images were collected with an sCMOS camera (Andor Zyla 5.5) without further magnification resulting—after 2 × 2 binning—in an effective pixel size of 130 nm per pixel. The system was controlled by the NIS-Elements (Nikon) software. Images were processed using the ImageJ2 software[36].

**DNA-PAINT imaging in cells**. FAs of cells were brought to focus using 488 nm excitation. For DNA-PAINT imaging, samples were imaged with 561 nm excitation wavelength and a laser power of 20 mW at the sample (power density: 1.24 kW/cm²); qPAINT measurements were performed with reduced laser power of 10 mW at the sample. Depending on the DNA docking sequence, the imager strand concentration was set between 250 pM and 2.5 nM; imaging was performed in the presence of an oxygen scavenging and triplet state quencher system consisting of a solution of 1× PCA (Stock 40× PCA solution; Sigma, 37580), 1× PCD (Stock 100× PCD solution; Sigma, P8279), and 1× Trolox (Stock 100× Trolox solution; Sigma, 238813) in 1× PBS + 500 mM NaCl. To experimentally mimic different molecular densities, CA-P1 and CA-P3 docking strands were mixed in different ratios (1:2–1:10) and subsequently added to fixed cells. Typically, 80,000 frames at 100 ms exposure time were acquired for NND imaging, and 160,000 frames at 100 ms for qPAINT measurements.

**Multiplexed imaging with Exchange-PAINT**. CA-P1 and BG-P3 (in case of three-target Exchange-PAINT CA-R1-, BG-R2-, and P3-conjugated 9EG7 antibody)[37] were diluted in 1× PBS containing 0.02 % Triton X-100 and added to PFA-fixed quadruple knockout cells expressing SNAP-kindlin and talin-Halo447, or Halo-kindlin and talin-SNAP447, respectively. The cells were then washed thoroughly with 1× PBS and imaged in two (or three) subsequent steps by DNA-PAINT SR microscopy. In the first step, SNAP-kindlin or Halo-kindlin was imaged using 2.5 nM Cy3b-P3 imager strand concentration (250 pM Cy3b-R2 for triple color experiments). After washing, 2.5 or 1 nM Cy3b-P1 (or 250 pM Cy3b-R1 for triple color experiments) was added to image talin-Halo447 or talin-SNAP447. For triple color experiments, an additional round of exchange was performed with Cy3b-P3 (1 nM) to label 9EG7-bound β1-integrin. To confirm that results are unaffected by the employed docking strand sequence, experiments were repeated using CA-P3 and BG-P1 docking strands in combination with the respective imager strands.

**DNA Origami self-assembly**. DNA origami structures were designed using the Picasso software[38]. The DNA origami self-assembly was performed in a reaction mix (Supplementary Table 3) containing 10 nM p7249 scaffold strand M13mp18 (tilibit nanosystems), 100 nM folding staples (Eurofins), 10 nM biotinylated staples (Eurofins), and 1 μM P3 docking strand in 1× TE buffer containing 12.5 mM MgCl₂. Subsequently, the DNA origami self-assembled in a thermocycler running the following cycling protocol: step 1, 80 °C for 5 min; step 2, immediate cool down to 60 °C; step 3, further cool down from 60 to 4 °C in steps of 1 °C per 3.21 min.

**PEG precipitation for DNA origami**. Polyethylene glycol (PEG) was used to decrease the solubility of origami and induce origami precipitation[39]. Origami solution in 1× TE buffer with 12.5 mM MgCl₂ was mixed 1:1 with 15% PEG buffer (7.5 g PEG-8000, 1× TAE, 12.5 mM MgCl₂, 500 mM NaCl) and centrifuged at 20,000 × g at 4 °C for 30 min. The supernatant was removed and origami was resuspended in folding buffer (12.5 mM MgCl₂, 10 mM Tris, 1 mM EDTA at pH

8.0). Centrifugation and supernatant removal were repeated three times. Origami was stored at −20 °C.

**Cell experiments with DNA origami**. Cells were seeded, fixed, and labeled as described above. To perform qPAINT experiments, labeling solution was removed and cells were washed 3× times with 1× PBS. Next, 200 μl BSA-Biotin solution (1 mg/ml BSA-Biotin in buffer A+ (10 mM Tris-HCl, 100 mM NaCl, and 0.05% Tween-20, pH 8.0)) was added and incubated for 10 min. The dish was then carefully washed with buffer A+, 200 μl streptavidin solution (0.5 mg/ml in buffer A+) was added, and incubated for another 10 min. Afterwards, the dish was washed with buffer A+ and subsequently with buffer C (1× PBS + 500 mM NaCl). Then, 200 μl of biotin-labeled DNA origami solution was added (200 pM in buffer C) and incubated for 60 min. Finally, the dish was carefully washed with buffer C and imaging buffer was added.

**Antibody conjugation to DNA-PAINT docking strands**. The integrin β1 9EG7 antibody was conjugated to DNA-PAINT docking strands using a bifunctional NHS ester crosslinker harboring an additional trans-cyclooctene moiety (TCO; TCO-NHS ester ((E)cyclooct-4-enyl-2,5-dioxo-1-pyrrolidinyl carbonate), Jena Bioscience, CLK-1016-25), which was later reacted with a methyltetrazine-PEG5-modified DNA strand to yield the final antibody–DNA conjugate[40]. In brief, the antibody storage buffer was exchanged via dialysis to 1× PBS overnight at 4 °C under constant stirring. The antibody was then concentrated with 100 kDa Amicon spin filters (Merck/EMD Millipore, UFC500396), TCO-NHS ester crosslinker was added at 10× molar excess and incubated for 2 h at 4 °C on a shaker. Afterwards, 7k zeba spin desalting columns (Thermo Fisher, 89882) were used to remove unreacted crosslinker. Tz-DNA was added to the purified antibody-crosslinker solution at 5× molar excess and incubated for 1 h at room temperature. Subsequently, amicon spin filters were used to remove free Tz-DNA and the antibody conjugate was stored at 4 °C.

**NB conjugation to DNA-PAINT docking strands**. GFP-NB was conjugated site-specifically to DNA-PAINT docking strands by using a bifunctional crosslinker, harboring a maleimide for attachment to the cysteine residue on the NB as well as a DBCO click-chemistry group for attachment to azide-modified DNA[5,38,41]. In brief, nanobodies were concentrated via 10 kDa amicon spin filters and buffer exchanged to 5 mM TCEP ((tris(2-carboxyethyl)phosphine)) + 1× PBS at pH 7.0. For the reduction of disulfide bonds, 5 mM TCEP was added to the GFP-NB and incubated for 2 h in the dark at 4 °C. Subsequently, TCEP was removed by 10 kDa amicon spin filters and buffer was exchanged to 1× PBS. Next, 20× molar excess of DBCO-maleimide crosslinker was added in a volume of 5 μl to the NB solution and incubated overnight at 4 °C on a shaker. Afterwards, free DBCO-maleimide crosslinker was removed via 10 kDa amicon spin filters and followed by a copper-free click reaction with an azide-modified DNA. For that, azide-DNA was added at 10× molar excess directly to the GFP-NB crosslinker solution and incubated for 1 h at room temperature. A buffer exchange using 10 kDa amicon spin filters to 1× PBS was performed and GFP-NB-DNA was purified from free excess DNA using anion-ion-exchange chromatography with a HiTrap Q column and a salt gradient from 1× PBS to 1× PBS + 1 M NaCl over the course of 30 min. Peak fractions were rebuffered and concentrated via 10 kDa amicon spin filters into 1× PBS.

**Image reconstruction**. Images were reconstructed with the Picasso software (latest version available on https://github.com/jungmannlab/picasso). Drift correction was performed stepwise starting with the gold nanoparticles for global drift correction, followed by image sub-stack cross-correlation analysis. Localization precision was determined by nearest-neighbor-based analysis[20].

**qPAINT analysis**. First, images were localized and drift corrected[38] and single binding sites on DNA origami structures were manually selected using the "Pick" tool in the Picasso Render module (~200–500 single origami binding sites per image). Afterwards, the selected binding sites were calibrated to one unit per binding site and the influx rate was estimated from the binding kinetics of the selected single binding sites on DNA origami[21,38]. The binding kinetics depend on the imager strand length, GC content, buffer salt concentration, and imager concentration. Talin localization clouds were selected (~1000–1500 picks/image) and the mean number of binding sites per selected localization cloud in FAs were calculated from the calibrated influx rate.

**Data processing and analysis**. For further analysis, we used DBSCAN as a data clustering algorithm[38,42]. DBSCAN detects localization clouds by looking for minimal numbers of localizations within a circle with the radius ε. Moreover, the algorithm utilizes a minimum number of points within an area of the circle as a second parameter. For ε, we used the localization precision in pixels of our images (nearest-neighbor-based analysis) and a minimum number of points were chosen according to the binding frequency of the imager strand and experiments used the parameters detailed in Supplementary Tables 4 and 5. Furthermore, we implemented a mean frame filter and a standard deviation filter to remove unspecific signals of the imager strands. In brief, repetitive transient binding to single sites in

the cells leads to a mean frame of roughly half the number of total frames in the acquisition window. To remove DBSCAN detected localization clouds, which are not continuously visited by an imager strand over the whole course of image acquisition, we fitted the mean frame value of all detected localization clouds and the cut-off value was set at plus or minus the standard deviation. Further sticking events, which were not removed by mean frame filtering due to sticking events occurring in the range of the mean frame filter, we used a standard deviation frame filter to remove long binding events (imager sticking) indicating unspecific binding events. These non-repetitive long binding events will lead to a mean frame located within the frames of their random appearance distributed in the acquisition window and thus resulting in a small standard deviation of the mean frame (Supplementary Table 5). For NND calculations, we used custom-written python scripts based on k-d tree analysis[43] to calculate the nearest neighbor within the dataset. For colocalization analysis, the NND for each molecule of one dataset with respect to the reference dataset was calculated (K2T, I2T, and I2K). Molecular densities were calculated by dividing the determined number of molecules within the FA mask by the respective FA mask area. The visualization is based on the convex hull of DBSCAN detected, grouped localization clouds. Note that the overlap of the colored regions depends on the chosen intensity, which does not affect the complex formation and distance analysis calculation.

**Fitting of NND distributions**. Plotting the NND over a logarithmic distance scale results in symmetric, Gaussian-shaped distributions pointing towards a structural order parameter. The simplest assumption of order is a random point localization, which—in two dimensions—is mathematically described by the 2D Poisson point process and its respective homogeneous Poisson density probability function ($\rho$ being the density), as described in equation (1):

$$P(r) = 2\pi r \rho e^{-\pi \rho r^2}$$

NND data were fitted in Origin9.1 using this custom-built fitting function.

**Simulation parameters**. Simulations of random particle distributions were performed with custom python scripts; parameters for DNA-PAINT simulations are detailed in Supplementary Table 6. In brief, for random distributions, random $x$- and $y$-coordinates were generated using the molecular particle densities extracted from the measured DNA-PAINT data. In agreement with previously published data[4,5] and our own talin-CalC experiments, we assumed LEs of 30% (HaloTag) and 20% (SNAP-tag). DNA-PAINT simulations were performed with the previously reported Picasso software[38] using parameters that were extracted from our experimental data to mimic raw data for image reconstruction and postprocessing filter steps (shown in Supplementary Table 6). To estimate the absolute molecular density and the corresponding NND, we set the HaloTag LE to 30%. To determine the localization cloud DE of the DBSCAN analysis, we compared manual selected localization cloud data with automatically analyzed data. Assuming an optimal DE of 100% for the manually selected data, a DE of ~50–60% for the DBSCAN analysis was estimated.

For simulations of complex formation, single sites of two protein species were simulated with the estimated final mean molecular density of 611 particles/μm². After randomly distributing the molecules in the first step, randomly chosen protein positions of the second species were reassigned to form different degrees of complex formation (0–100%) with the first species. Distance of a complex was set to be 12–16 nm (from 0 to 100% complex formation), which corresponds to the measured distances in the CalC control construct. Afterwards, 70% (HaloTag) and 80% (SNAP-tag) of protein positions were randomly removed to account for the LE. The remaining positions of the two proteins were used to calculate the degree of colocalization within 25 nm per complex formation. To estimate how the sensitivity towards protein complex formation depends on the molecular density, different densities covering a range of 100–1000 molecules/μm² with constant LE were simulated. To test the effects of LE, we assumed a constant molecular density of 611 particles/μm² and varied the LE for HaloTag (20–100%) and SNAP-tag (13–66%). To calculate the degree of colocalization within three random distributions, localization clouds were simulated with the measured molecular density of the respective experiment. All simulation scripts are available upon request.

**Estimation of the absolute molecular density of talin-1 in FAs**. To determine the absolute molecular density of talin-Halo447 in FAs, we considered three cases: best, worst, and intermediate LE and DE. The starting value of the molecular density, 99.25 ± 17.6 molecules/μm², has been extracted from the experimental data (talin-Halo447 at 16 h timepoint). As a lower bound, we assumed a LE of 25% and a DE of 33% with a molecular density of 82.4 molecules/μm², resulting in a density of 988 molecules/μm². For the intermediate scenario, a LE of 33% and DE of 50% with 99.25 molecules/μm² were assumed resulting in 600 molecules/μm². Finally, for the best-case scenario, a LE of 60% and DE of 80% with 117.6 molecules/μm² in FAs was assumed, resulting in 245 molecules/μm² after extrapolation. Calculating mean molecular density and its related sigma value results in an absolute molecular density of 611 ± 303 molecules/μm².

**Statistical analysis**. To determine if two data sets are equal, two-sample $t$-test has been used with a $p$-value < 0.05 being statistically significant. To determine the goodness of fit between experimental data and simulated data, a 2D Kolmogorov–Smirnov test was used[44]. Bootstrapping was performed to calculate the mean differences and standard deviation. In brief, 1000 data points (tuples) were randomly sampled out of each dataset consisting of ~30,000 tuples. This procedure was applied on both data sets and the maximum difference obtained by performing Kolmogorov–Smirnov test. This process was repeated 1000 times to obtain a series of maximum differences between the two data sets yielding mean differences and standard deviation. Simulated vs. simulated data and experimental vs. experimental data were compared as a control, resulting in high $p$-values for intrinsic data. Then, experimental data (integrin-kindlin-talin distances; I2KT) vs. randomly simulated data were compared, leading to high differences and thus low $p$-values (n.s., $p > 0.05$; *$p \leq 0.05$; **$p \leq 0.01$; ***$p \leq 0.001$).

**Reporting summary**. Further information on research design is available in the Nature Research Reporting Summary linked to this article.

## Data availability
The data supporting the findings of this study are available within the article and its Supplementary information. Any other relevant data are available upon reasonable request. Source data are provided with this paper.

## Code availability
The Picasso software can be accessed at: https://github.com/jungmannlab/picasso[38]. Any other code is available upon request.

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

## Acknowledgements

We thank Roland Wedlich-Söldner, Thomas Zobel, and the Imaging Network at the WWU Münster for excellent microscopy support, Sebastian Potthoff and the IT service of the ZIV (WWU Münster) for help with the data analysis infrastructure. We also thank Christoph Schreiber (LMU Munich, Department Prof. Rädler) for manufacturing and providing the micropatterned substrates, Ina Kowsky for generating cell lines, Anna-Lena Schweizer for helpful suggestions during data analysis, and Caroline Lindner for support with data analysis. This work was supported by grants from the German Research Council to C.G. (DFG, GR3399/6-1 and Inst 211/867-1). R.J. acknowledges support by the German Research Foundation through the Emmy Noether Program (DFG JU 2957/1-1), the SFB1032 (project A11), the European Research Council through an ERC Starting Grant (MolMap; grant agreement number 680241), the Allen Distinguished Investigator Program through The Paul G. Allen Frontiers Group, the Danish National Research Foundation (Centre for Cellular Signal Patterns, DNRF135), the Max Planck Foundation, and the Max Planck Society. T.S. acknowledges support by the DFG through the QBM graduate school.

## Author contributions

L.S.F. conceived and performed labeling strategies, experiments, and analyzed data. C.K. initialized the experimental studies, established protocols, developed the analysis pipeline, performed experiments, and analyzed data. T.S. conceived and performed labeling strategies, experiments, and analyzed data. M.T.S. developed the software. R.B. generated a cell line. R.F., R.J., and C.G. conceived the general idea. R.J. and C.G. supervised the study, and C.G. wrote the paper with the input from L.S.F., C.K., T.S., and R.J.

## Funding

## Competing interests

The authors declare no competing interests.
