## [Peer Review File · Nature Communications]

Reviewers' Comments:

Reviewer #1:

Remarks to the Author:

Single protein imaging approaches provide a useful complement to biochemical techniques in order to determine molecular mechanisms. However, incomplete labelling and detection currently hamper the application of the technology. In this manuscript, the authors describe a super-resolution workflow (qSMCL) to quantify protein levels in space. They apply the workflow to analyses of talin-kindlin and integrin-talin-kindlin associations within and outside focal adhesions.

I see little to criticize in the technical aspects of this study. The design of the qSMCL approach is clever, and all key controls appear to have been incorporated. The insights into regulation of integrin activation at focal adhesions are minimal (essentially confirming that talin and kindlin act in very close proximity during the process), but nonetheless will be useful to the field. Above all, the manuscript forms a basis for future studies requiring a quantitative understanding of molecular proximity and it will therefore be a valuable resource for anyone undertaking super-resolution imaging strategies. I support its acceptance.

Reviewer #2:

Remarks to the Author:

In the manuscript "Quantitative single-protein imaging reveals molecular complex formation of integrin, talin, and kindlin during cell adhesion" Fischer et al describe an approach and its application to quantify protein association in cell adhesion complexes.

The manuscript is interesting and the methodology that is developed has potential. However, as currently presented, I find that the presentation lacks clarity in a number of important places and the stringent requirements for this analysis approach to be applicable to a given protein system are not clearly pointed out.

1. Consistently I had to carefully read and consult the methods section to clarify points not clearly presented in the results section. The results section and figure legends need to explicitly spell out and mention details when relevant rather than force the reader to have to carefully distil details out of methods descriptions; essentially, vital info is not presented at the point where the reader needs to understand what was done. Some degree of repetition between results section and methods section is entirely ok and actually required to ensure readers can follow the arguments with more ease.

A. For example, the results section describing the first figure makes reference to 'signals' and also to 'point clouds' when it is not at all precisely defined what these terms mean: "allowed us to distinguish signals as close as 10–15 nm"; "talin-1 molecules per localization signal"; "detected similar values for DNA origami and talin-1 localization clouds".

The word 'signal' is far too general to be able to work out what is meant; what is a "localization signal"? Then the terminology switches to "point clouds"; it took reading the methods to infer that what appears to be referred to are groups of localisation events ("points") obtained by a process termed 'picking' which is a term adopted in the software used to select a subset of localisation events, it would seem. Here I adopt the term 'localisation event' to mean a single molecule localisation in a single frame of the raw data underlying the analysis, something the authors appear to refer to as "points".

From looking at the rendered image (Fig. 1c) it seems that the rendered staining is punctate in appearance and by inspection of the methods that the authors used the 'picking' method to select localisation events underlying individual staining puncta in images such as the one shown in Fig.

1c; it is not clear how "picking" works from the manuscript, e.g. does the user draw a box around a punctum in the image? This may well be described in other references but it was not obvious and a brief summary how picking works and its ability to segment events between nearby puncta needs to be pointed out in this MS. Is it purely down to the operator how "points" are selected or does the software use an algorithm to find events in the vicinity of where the user clicked/drew?

B. "We then correlated the binding kinetics of single sites associated with DNA origami and talin-Halo447 to convert the observed binding frequency to absolute numbers of binding sites, and detected similar values for DNA origami and talin-1 localization clouds"; this seemed confusing when first read, how were binding kinetics correlated? Consulting the methods it was much clearer what was done, namely the known number of binding sites in DNA origami (a single site per resolved punctum in the DNA origami images) was used to calibrate the qPAINT method to obtain the influx rate, and this calibration was then used to obtain the number of binding sites associated with the (currently not clearly defined) "localisation clouds" associated with puncta of talin-1 appearance in the DNA-PAINT super resolution images. The authors should use this much clearer description from the methods in the main text to say what was done.

C. The results section refers several times to labelling efficiency (LE) and detection efficiency (DE) but it took again consulting the methods section to reveal what actual values were used. From the methods it seemed that values of 30 % (HaloTag) and 20 % (SNAP-tag) were used, and detection efficiency of the DBCAN method was estimated as 50-60%.

In relation to this point, from the results text description and legend it was not clear how the drawn points in Fig. 2i were obtained, both points marked as 'experiment' and 'simulation'. I guess that the labelling efficiency and detection efficiency were varied in the simulation and (and experiments?), that the best guess value is then taken from the simulation as the extrapolation to 100% LE and DE, although this is nowhere clearly spelled out. Similarly, the experimental points are measurements taken at different time points? Explanations for how the simulation points and the experimental points were obtained need to be added. If experimental points are from different times this should be indicated in the plot (e.g. by colour coding).

2. The stringent requirements/limitations of the approach that the authors are presenting here is not clearly communicated. It seems that for the method to be applicable:

- detection and segmentation of single protein "point clouds" are required; this is clearly not always possible
- a qPAINT calibration has to be available (shown nicely with the origami here)
- a 1:1 labelling (or known precise ratio) of proteins with docking sites needs to be achieved
- estimates of labelling and detection efficiency need to be obtained

The authors need to make these requirements more explicit, ideally listing them in the discussion (which is currently extremely brief) clearly and discuss their applicability to different biological imaging scenarios and imaging modalities.

A. Resolution: it seems that imaging resolution must be high enough to not merge point clouds of individual proteins; in the application shown in this MS the optical imaging conditions are probably near optimal: nearly 2D structures right at the coverslip surface. By contrast, deep within the nucleus of a cell with 3D localisation required this level of resolution may generally not be achievable, at least not with current widely available super-res modalities.

In connection with this point, the authors state "This procedure allowed us to distinguish signals as close as 10–15 nm". The authors should present actual resolution estimates, using FRC analysis, taking care to avoid spurious correlations from imager events lasting over many frames.

B. Labeling: the SNAP/Halo tag labelling is a good example of a 1:1 protein label (at least for single subunit proteins); interestingly, for the triple labelling shown in Fig. 4 the authors did not

carry out the same level of analysis as carried out for talin/kindlin. Is this due to limitations of AB based labelling of integrin?

C. labeling/detection efficiencies;

Labeling: from the data present in this MS, it seems that only the relative LE of Snap and Halo tags were directly measured, rather than absolute (Snap 30% less efficient than Halo); the absolute numbers seem to be based on measurements with NPCs in previous papers; is it clear that SNAP and Halo-tag LEs are independent of the protein target these are fused to? To this reviewer it seems a priori not clear that this is the case, e.g. due to different molecular accessibility etc.

In connection with this point, LE and DE estimates will also have uncertainties; the authors should add information how sensitively their protein number and complex formation fractions etc depend on these estimates.

Detection: DBSCAN detection efficiencies were determined with picking as comparison; what evidence is there that picking detects 100% of "point clouds"? From extended data Fig. 2g it seems that DBSCAN may also not be able to always segment perfectly, leading to possibly merged point clouds representing several proteins; did this never occur? If it did, how were these point clouds rejected?

In my own experience with DBSCAN its performance strongly depends on the parameter choice. The methods mention how epsilon was determined, however the choice of MinPts is less clear: "MinPts were chosen according to the binding frequency of the imager strand." Can you be more precise how MinPts were chosen, what values were actually used? How variable were these values for different data sets? Is there an objective procedure to choose MinPts?

Point cloud display: DBSCAN testing and other panels (e.g. extended data figure 3c, fig 3b, fig 4c) show uniformly coloured areas presumably supposed to represent "point clouds"; it is unclear how these are actually related to the underlying point clouds; is it related to the convex hull of the underlying point clouds? Please clearly describe how visualisations were constructed. It seemed surprising that the coloured regions actually show overlap, see extended data figure 3c.

Minor:

"Localization precision was determined by nearest-neighbor based analysis (NeNA)²⁸." This statement seems to refer to ref 20, not 28.

Reviewer #3:

Remarks to the Author:

Determining the absolute number of molecules in a protein complex is not easy. Instead, a set of stochastic single-molecule event data reveal the same result with ensemble data. For example, it has been suggested that a certain number of single-molecule events can determine the boundaries of the complexes in superresolution images (Patterson et al. 2010).

The author's main claim is to precisely determine the absolute molecular densities of talin-1, kindlin and integrin in FA using the coupled qSMCL + DNA-PAINT technique, which has not been reported yet in this field. In principle, the authors used the calibrated binding kinetics between DNA origami nanostructures called "docking strands" and "imager strands" labeled by dye or cy3 respectively, which was simultaneously imaged next to the cells expressing talin-halo tag and targeted with same soluble "imager strands" that penetrated the cell membrane. Then, based on Poisson process measurement of tali-1 spatial distribution, the author performed random simulations for clustering to validate the experimentally determined density and spatial distribution

length, called NND.

The combined technique applied to this issue is unique because it allows the system to be controlled using previously proven DNA origami engineering. However, it is still unclear to me whether the number of talin-1 and kindlin-2 in FA is absolute or not, at this point. The authors assumed that the binding kinetics between DAN-PAINT docking strands and DNA-imager strands are the same in solution and fixed cell cytoplasm, which may differ in terms of "binding time", which subsequently affects the binding frequency to the same binding site. Viscosity, concentration and physical hindrance should be considered in solution versus the cell membrane that is critically change binding kinetics. One way to validate this problem is for the authors to compare the dwell time (photobleaching time not photo blinking event) of DNA-imager strands in origami and the cell-membrane.

Detail questions:

Result 1: Determine absolute # of talin-1 in FA.

1) The authors convert the observed time-trace of the binding frequency to the absolute numbers of binding sites. In detail (not described in the manuscript, but referenced to previous papers), the authors used the mean off-time (τ_d^*) of cy3 or the dye tagged to DNA-imager to determine the binding site number. τ_d^* was derived from the distribution of the waiting-time (fluorescent off time) in the time-trace of the binding frequency. Another parameter needed to calculate the binding site number is the influx rate that is equal to $k_{on} \times [\text{imager concentration}]$. Then finally, the number of binding sites is equal to $(\tau_d^* \times k_{on} \times [\text{imager concentration}])^{-1}$. In this equation, the author must prove that the k_{on} and [DNA-imager concentration] of solution and cell cytoplasm are the same. In general, the binding kinetics of proteins in cellular systems is complex and affected by not only the concentration of reactants, but also geometric factors and the presence of competitors. In addition, the rate of diffusion of DAN-imager in the fixed cell cytoplasm may differ from that of the medium, since it is another critical factor in protein-protein binding frequency. The authors should discuss this issue.

2) How does the author determine influx rate? They used DNA-imager between 250 pM and 2.5 nM. Was k_{on} determined in this system?

3) Increasing the observation time increases the total number of binding events due to the irreversible binding/unbinding kinetics between the Halo tagged-Talin and DNA-imager strands. How does the author consider the rebinding events at the previous binding site, which would overcount the absolute number of binding sites?

4) How do the authors discriminate between one or multiple binding sites based on the digitized fluorescence signal? For example, in Figures 1 J and K, both single and triple bonds show the same on/off digitized fluorescence signal. Is the intensity for triple binding 3-time higher than for the single bonded fluorescence signal (it looks like) ? Or does the # of events contribute to fluorescence signal intensity (Figure 1 k is 3-time higher than at Figure 1 J (also it looks like)? When authors show the raw data of the intensity profile for the event traces, the audience can easily understand the processing protocol instead of just showing only the digitized fluorescence intensity trace.

Result 2: Cluster analysis and theoretical simulation

1) How do the authors eliminate unspecified single molecule events when using DBSCAN to detect clusters? The criteria used by authors should be explained along with their reasoning, since every single-molecule event on the screen is a real event. This issue is important while thresholding molecular events in SMLM images and in sptPALM signal. For example, in a living cell system, a signal for one- frame at a high video rate (10 - 30 fps) can be considered as a non-specific event and filtered out. Indeed, in principle, the portion of single frame events is significant in fluorescent labeled imaging data, and it is difficult to estimate how many of such events are involved in the total data of the fixed sample. Instead, the authors used mean frame and std frame filtering when removing non-specific events, which are tail parts in frame distributions (Extended data figure 2 h-1). Are the distribution of mean frame and std frame the same in solution (i.e., DNA origami control binding) and in cell cytoplasm (talin-halo:DNA-imager)?

2) How did the author perform the segmentation process shown in the extended data Figure 2g-i?

Unlike DBSCAN control origami data (Extended data figure 3), there are large clusters, some of which are interconnected (Extended data figure 2 g). It is a good idea to briefly discuss the automatic cluster identification algorithm.

3) Based on the results of this section, the authors concluded that the spatial distribution of talin-1 in FA is governed by the molecular density instead of the molecular ruler function of talin-1 proposed previously. They calculated the density of talin-1 as 600 molecules/ μm^2 in FA, which "completely" packed in the adhesion (Talin-1 is 270kDa, which is 10x larger than GFP, assuming even a linear structure of 4.8 nm width and 420 nm length). Thus, this excludes the existence of other molecules which is normally un-realistic because molecule-molecule interactions occur in a mixture of different proteins as shown in figures 3 and 4 (overlapping between talin-1, kindlin, and integrin $\beta 1$). Is this artifact caused by a reversible binding event at the same binding site as mentioned above? This issue is related to the following questions.

Result 3: Double labelling imaging of Kindlin-2 and Talin shows significant overlap in the clusters shown in Figure 3. In this case, is the absolute number of each molecule the same as determined in the single-color imaging data shown in Figure 1?

Result 4: Again, in the triple DNA-PAINT imaging of talin, kindlin and integrin $\beta 1$, the mutual overlapped areas between clusters are significant. Can the author further analyze the absolute number of copies of each component in this overlapped area? Since this paper's primary claim relates to this issue (determining absolute number of molecules in FA), this question will be of interesting to the audience. The lateral distribution of these three components is not a big deal as the 3D spatial information is still missing in the 2-D projection images.

Overall, it is a good idea to consider publishing this paper if the authors can address some of the critical issues raised above. This is because experimentalists generally use relative comparisons rather than focusing on the absolute numbers due to many unresolved factors, especially in protein-protein kinetics in cells (Oh et al. 2012). And in vivo and fixed samples may have very different properties that affect binding kinetics and consequently influence measurements of the absolute copy number.

Oh, D., M. Ogiue-Ikeda, J. A. Jadwin, K. Machida, B. J. Mayer, and J. Yu. 2012. 'Fast rebinding increases dwell time of Src homology 2 (SH2)-containing proteins near the plasma membrane', *Proc Natl Acad Sci U S A*, 109: 14024-9.

Patterson, G., M. Davidson, S. Manley, and J. Lippincott-Schwartz. 2010. 'Superresolution imaging using single-molecule localization', *Annu Rev Phys Chem*, 61: 345-67.

Quantitative single-protein imaging reveals molecular assembly of integrin, talin, and kindlin during cell adhesion

Lisa S. Fischer, Christoph Klingner, Thomas Schlichthaerle, Maximilian T. Strauss, Ralph Böttcher, Reinhard Fässler, Ralf Jungmann and Carsten Grashoff

Response to Reviewer #1

Single protein imaging approaches provide a useful complement to biochemical techniques in order to determine molecular mechanisms. However, incomplete labelling and detection currently hamper the application of the technology. In this manuscript, the authors describe a super-resolution workflow (qSMCL) to quantify protein levels in space. They apply the workflow to analyses of talin-kindlin and integrin-talin-kindlin associations within and outside focal adhesions.

I see little to criticize in the technical aspects of this study. The design of the qSMCL approach is clever, and all key controls appear to have been incorporated. The insights into regulation of integrin activation at focal adhesions are minimal (essentially confirming that talin and kindlin act in very close proximity during the process), but nonetheless will be useful to the field. Above all, the manuscript forms a basis for future studies requiring a quantitative understanding of molecular proximity and it will therefore be a valuable resource for anyone undertaking super-resolution imaging strategies. I support its acceptance.

Response: We thank the reviewer for evaluating our manuscript and the positive comments. In response to the remarks of the other two referees, we have included additional information into the main text and the methods section, and we prepared additional Supplementary Figures and Supplementary Tables to explain data analysis procedures in more detail. We believe that this has further strengthened the manuscript.

Response to Reviewer #2

In the manuscript “Quantitative single-protein imaging reveals molecular complex formation of integrin, talin, and kindlin during cell adhesion” Fischer et al describe an approach and its application to quantify protein association in cell adhesion complexes.

The manuscript is interesting and the methodology that is developed has potential. However, as currently presented, I find that the presentation lacks clarity in a number of important places and the stringent requirements for this analysis approach to be applicable to a given protein system are not clearly pointed out.

1. Consistently I had to carefully read and consult the methods section to clarify points not clearly presented in the results section. The results section and figure legends need to explicitly spell out and mention details when relevant rather than force the reader to have to carefully distil details out of methods descriptions; essentially, vital info is not presented at the point where the reader needs to understand what was done. Some degree of repetition between results section and methods section is entirely ok and actually required to ensure readers can follow the arguments with more ease.

Response: We thank the reviewer for the positive remarks and the very constructive suggestions. As requested, we have made a number of adjustments in the main text and the methods section to clarify experimental details. We also include additional Extended Data Figures (now called Supplementary Figures) and two additional Supplementary Tables. Together, this should allow the reader to better follow the experimental procedures, and we hope that the reviewer will now be able to support the publication of this study.

A. For example, the results section describing the first figure makes reference to ‘signals’ and also to ‘point clouds’ when it is not at all precisely defined what these terms mean: “allowed us to distinguish signals as close as 10–15 nm”; “talin-1 molecules per localization signal”; “detected similar values for DNA origami and talin-1 localization clouds”.

The word ‘signal’ is far too general to be able to work out what is meant; what is a “localization signal”? Then the terminology switches to “point clouds”; it took reading the methods to infer that what appears to be referred to are groups of localisation events (“points”) obtained by a process termed ‘picking’ which is a term adopted in the software used to select a subset of localisation events, it would seem. Here I adopt the term ‘localisation event’ to mean a single molecule localisation in a single frame of the raw data underlying the analysis, something the authors appear to refer to as “points”.

Response: We thank the reviewer for this feedback and we agree that the terminology was confusing. We clarified this by introducing the term *localization cloud* at the initial stage of the manuscript, as long as it is not yet established that we achieve molecular resolution. Once this is demonstrated by the qPAINT experiments in Fig. 1e-o, we use the term *molecules* (i.e. talin-1 molecules, kindlin-2 molecules or integrin molecules) but no longer speak of *cloud*, *signal* or *event*. To be as clear as possible, we generated an additional Extended Data Figure (now Supplementary

Fig. 1a), in which the terminology is explained. We hope that this will make it easier for the reader to follow the experiments.

From looking at the rendered image (Fig. 1c) it seems that the rendered staining is punctate in appearance and by inspection of the methods that the authors used the ‘picking’ method to select localisation events underlying individual staining puncta in images such as the one shown in Fig. 1c; it is not clear how “picking” works from the manuscript, e.g. does the user draw a box around a punctum in the image? This may well be described in other references but it was not obvious and a brief summary how picking works and its ability to segment events between nearby puncta needs to be pointed out in this MS. Is it purely down to the operator how “points” are selected or does the software use an algorithm to find events in the vicinity of where the user clicked/drew?

Response: To clarify this, we now use the term *manual selection* instead of *picking*, and we explain in Supplementary Fig. 1b how the procedure works. As illustrated, *picks* are selected manually by the operator.

B. “We then correlated the binding kinetics of single sites associated with DNA origami and talin-Halo447 to convert the observed binding frequency to absolute numbers of binding sites, and detected similar values for DNA origami and talin-1 localization clouds”; this seemed confusing when first read, how were binding kinetics correlated? Consulting the methods it was much clearer what was done, namely the known number of binding sites in DNA origami (a single site per resolved punctum in the DNA origami images) was used to calibrate the qPAINT method to obtain the influx rate, and this calibration was then used to obtain the number of binding sites associated with the (currently not clearly defined) “localisation clouds” associated with puncta of talin-1 appearance in the DNA-PAINT super resolution images. The authors should use this much clearer description from the methods in the main text to say what was done.

Response: We changed the paragraph in the main text according to the reviewer’s suggestion and also explain the qPAINT calibration in Supplementary Fig. 1b.

The text now reads: “*To quantify the exact number of talin-1 molecules per localization cloud, we implemented quantitative-PAINT (qPAINT) analyses²¹ by placing DNA origami nanostructures with a defined number of docking strands next to talin-Halo447 expressing cells (Fig. 1e-g). After image acquisition, we used the observed binding kinetics to estimate the imager strand influx rate on single binding sites associated with DNA origami, and utilized this calibration to determine the number of binding sites in talin localization clouds^{21,34} (Supplementary Fig. 1b). This analysis revealed similar values for the single-binding sites on DNA origami and talin-1 localization clouds (DNA origami: 1 ± 0.3 binding sites; talin-1: 1 ± 0.4 binding sites; Fig. 1h, i) indicating that individual talin-1 molecules are detected (Supplementary Fig. 3).*”

We believe that this description will make it easier to follow the experimental procedure and we thank the reviewer for the suggestion.

C. The results section refers several times to labelling efficiency (LE) and detection efficiency (DE) but it took again consulting the methods section to reveal what actual values were used. From the methods it seemed that values of 30 % (HaloTag) and 20 % (SNAP-tag) were used, and detection efficiency of the DBSCAN method was estimated as 50-60%.

Response: We thank the reviewer for this suggestion. The values were added into the main text, which now reads:” *According to a previous study⁴, the LE of HaloTag was set to 30 %, and the DE for the DBSCAN based analysis was estimated, based on a comparison with manually selected data, to be 50–60 % (Fig. 2d).*”

We repeat this information at a later stage in the manuscript, when also the LE of the SNAP-tag is considered: “*We thus established a theoretical framework that accounts for the experimentally observed protein distributions, molecular densities, and previously determined labelling efficiencies for HaloTag (30 %) and SNAP-tag (20 %) avoiding bias through the single-molecule localization process⁵.*”

In relation to this point, from the results text description and legend it was not clear how the drawn points in Fig. 2i were obtained, both points marked as ‘experiment’ and ‘simulation’. I guess that the labelling efficiency and detection efficiency were varied in the simulation and (and experiments?), that the best guess value is then taken from the simulation as the extrapolation to 100% LE and DE, although this is nowhere clearly spelled out. Similarly, the experimental points are measurements taken at different time points? Explanations for how the simulation points and the experimental points were obtained need to be added. If experimental points are from different times this should be indicated in the plot (e.g. by colour coding).

Response: We added a more detailed explanation in the main text to clarify this point.

The new section now reads: “*Finally, we estimated the absolute molecular density of talin 1 in FAs of cells 16 h after seeding (Fig. 2i). We first mimicked lower molecular densities by successively reducing the docking strand concentration in our experiments. We then confirmed that the resulting data set are described by a simulation using the experimentally obtained NNDs and the above-mentioned estimates of LE and DE. We then extrapolated the data by simulating a gradual increase of LE and DE. These simulations predicted an absolute talin-to-talin distance of 20–25 nm in mature FAs (16 h) with a molecular density of approximately 600 talin-1 molecules/ μm^2 (Fig. 2i).*”

2. The stringent requirements/limitations of the approach that the authors are presenting here is not clearly communicatd. It seems that for the method to be applicable:

- detection and segmentation of single protein “point clouds” are required; this is clearly not always possible*
- a qPAINT calibration has to be available (shown nicely with the origami here)*
- a 1:1 labelling (or known precise ratio) of proteins with docking sites needs to be achieved*

- estimates of labelling and detection efficiency need to be obtained

The authors need to make these requirements more explicit, ideally listing them in the discussion (which is currently extremely brief) clearly and discuss their applicability to different biological imaging scenarios and imaging modalities.

Response: We thank the reviewer for this suggestion and added these details into the discussion. The new segment reads: “*The approach should be adaptable to all SMLM techniques obtaining molecular resolution such as MINFLUX³, Expansion SMLM²⁹, and – as shown in this study – PAINT-based approaches. The experiments require careful calibration to determine the number of molecules per localization cloud, which can be realized by qPAINT measurements using DNA origami or, potentially, recently developed localization-based fluorescence correlation spectroscopy (IbFCS)³⁰. Furthermore, the theoretical simulations require an estimate of the labelling and detection efficiencies, and biological controls are key to validate the calculated degree of protein-protein association in the given subcellular context.*”

A. Resolution: it seems that imaging resolution must be high enough to not merge point clouds of individual proteins; in the application shown in this MS the optical imaging conditions are probably near optimal: nearly 2D structures right at the coverslip surface. By contrast, deep within the nucleus of a cell with 3D localisation required this level of resolution may generally not be achievable, at least not with current widely available super-res modalities.

Response: Please note that SMLM was already applied to the nuclear pore complex (NPC), where single copies of nucleoporins have been resolved in 3D (Schlichthaerle et al, *Angewandte Chemie*, 2019; Thevathasan et al, *Nature Methods*, 2019). We agree with the reviewer that such 3D measurements are technically more challenging as compared to 2D analyses, but they are certainly feasible and the here developed framework should be also useful for these approaches.

In connection with this point, the authors state “This procedure allowed us to distinguish signals as close as 10–15 nm”. The authors should present actual resolution estimates, using FRC analysis, taking care to avoid spurious correlations from imager events lasting over many frames.

Response: Please note that we performed a localization precision analysis based on the previously described NeNA metric (Endesfelder et al, *Histochemistry and cell biology*, 2014); these data are shown in Supplementary Fig. 2c. In response to the reviewer’s suggestion we also performed an FRC analysis (Nieuwenhuizen et al, *Nature Methods*, 2013), which indicated very similar values as compared to the NeNA analysis (please see Figure 1 for reviewer #2 below). We would also like to note that NeNA-based (e.g. localization precision based) estimates for resolution are more suitable for our data sets, as we indeed show that single protein molecules can be resolved under our imaging conditions. This allows us to calculate localization precisions (and thus resolution) directly from our “point clouds”. We furthermore would argue that this “digital molecular imaging” yields images of point patterns, where only localization precision dictates resolution and FRC-reported estimates might actually be prone to artefacts from point patterns.

Figure 1 for reviewer #2. Comparison of FRC and NeNA-based analysis a. The FRC curve indicates the decay of the correlation with spatial frequency. The curve drops below the threshold ($1/7 \approx 0.1428$) and thus resulting in a threshold value at $q = 0.1426 \text{ nm}^{-1}$, which is equivalent to 6.991 nm . **b.** The estimated localization precision by NeNA calculation yielded a similar value of 6.474 nm . Data were fitted using the least square method and best value was extracted (0.04978 px).

B. Labelling: the SNAP/Halo tag labelling is a good example of a 1:1 protein label (at least for single subunit proteins); interestingly, for the triple labelling shown in Fig. 4 the authors did not carry out the same level of analysis as carried out for talin/Kindlin. Is this due to limitations of AB based labelling of integrin?

Response: The reviewer is correct. The antibody was labelled using NHS chemistry, which yields a heterogenous mixture of DNA strands attached to the antibody; this prevents the quantification of absolute integrin numbers. To be more precise, we now state in the main text more precisely: “we conjugated DNA-PAINT ‘docking’ strands to the 9EG7 antibody”, and we include another sentence into the Figure legend stating that the approach does not allow the counting of integrin receptors in a single localization cloud.

Labeling: from the data present in this MS, it seems that only the relative LE of Snap and Halo tags were directly measured, rather than absolute (Snap 30% less efficient than Halo); the absolute numbers seem to be based on measurements with NPCs in previous papers; is it clear that SNAP and Halo-tag LEs are independent of the protein target these are fused to? To this reviewer it seems a priori not clear that this is the case, e.g. due to different molecular accessibility etc.

Response: The reviewer is correct in assuming that the absolute labelling efficiency was inferred from a previous manuscript (Schlichthaerle et al, *Angewandte Chemie*, 2019). To confirm this assumption, we compared the simulation of the perfect dimer, in which we assumed 30 % LE of HaloTag and 20 % LE for SNAP-tag (Fig. 3e), with experimental data of the CalC construct which comprises a HaloTag and a SNAP-tag and mimics a dimer. As shown in Fig. 3h, simulation and experiments correlate very well suggesting that the assumption on LEs are reasonable.

In connection with this point, LE and DE estimates will also have uncertainties; the authors should add information how sensitively their protein number and complex formation fractions etc depend on these estimates.

Response: We thank the reviewer for this suggestion. We now include further simulations into the manuscript showing how complex formation depends on the molecular densities (new Supplementary Fig. 10a) and labelling efficiencies (new Supplementary Fig. 10b).

Detection: DBSCAN detection efficiencies were determined with picking as comparison; what evidence is there that picking detects 100% of “point clouds”? From extended data Fig. 2g it seems that DBSCAN may also not be able to always segment perfectly, leading to possibly merged point clouds representing several proteins; did this never occur? If it did, how were these point clouds rejected?

Response: As now indicated in Supplementary Fig. 1b, *picking* is a manual selection process. The procedure uses the ‘Pick’ tool of the Picasso software (<https://github.com/jungmannlab/picasso>) and is very laborious; however, it has a detection efficiency of 100 %. The data analysis pipeline uses three consecutive DBSCAN steps, which is now indicated more clearly in Supplementary Fig. 4 (former Extended Data Fig. 2). The first two rounds of DBSCAN generate a cluster file for ‘mean_frame’ and ‘std_frame’ filtering using a slightly larger radius. In the third DBSCAN, a smaller radius determined by the localization precision (\sim NeNA) is applied to avoid merging. We cannot entirely eliminate merging events by this procedure but we can reduce these unwanted effects significantly. We modified Supplementary Fig. 4 (former Extended Data Fig. 2) to illustrate this point more efficiently.

In my own experience with DBSCAN its performance strongly depends on the parameter choice. The methods mention how epsilon was determined, however the choice of MinPts is less clear: “MinPts were chosen according to the binding frequency of the imager strand.” Can you be more precise how MinPts were chosen, what values were actually used? How variable were these values for different data sets? Is there an objective procedure to choose MinPts?

Response: We agree with the reviewer that MinPts is an important parameter and we included another table into the Supplementary data, where all relevant parameters are shown. In our experience, the optimal MinPts value strongly correlates with protein expression levels and imager concentration, and are ideally determined manually by analyzing the binding frequency on single localization clouds. Since we worked with reconstituted knockout cell lines and highly similar expression levels, we could keep the imager concentration constant and thus use the same parameter set for all experiments imaged with the same DNA-strand. On average, we observed \sim 120 localizations per localization cloud and this value was virtually unchanged over three years of imaging (see Figure 2 for reviewer #2).

Figure 2 for reviewer #2. qPAINT reproducibility a-c.

Histograms of localizations per localization cloud reveal unimodal distributions and similar numbers of localizations per localization cloud over three years of data acquisition. **d.** Boxplots of localization per localization cloud validate similar mean values for data sets imaged over three years (median₂₀₁₈ = 128; median₂₀₁₉ = 101; median₂₀₂₀ = 127). Boxplots show median and 25th and 75th percentage with whiskers reaching to the last data point within 1.5x interquartile range.

Point cloud display: DBSCAN

testing and other panels (e.g. extended data figure 3c, fig 3b, fig 4c) show uniformly coloured areas presumably supposed to represent “point clouds”; it is unclear how these are actually related to the underlying point clouds; is it related to the convex hull of the underlying point clouds? Please clearly describe how visualisations were constructed. It seemed surprising that the coloured regions actually show overlap, see extended data figure 3c.

Response: Yes, we used the convex hull to visualize the shape of DBSCAN-detected localization clouds. The overlap of the coloured regions, as seen in Supplementary Fig. 5c (former Extended Data Fig. 3c), depends on the chosen intensity. Please note that this is just a way of visualization (see Fig. 3 for reviewer #2), which does not affect the complex formation or distance analysis calculation.

Figure 3 for reviewer #2. **a.** Individual segmented localization clouds on a 30 nm DNA origami detected with the DBSCAN procedure show overlapping colored regions. **b.** The same DNA origami data set showing individual localizations. **c.** Using a different intensity level for the same DBSCAN rendered image shows less signal overlap. **d.** Single colored image using identical intensity values as in (c). Scale bars: 20 nm (a-d).

“Localization precision was determined by nearest-neighbor based analysis (NeNA)28.” This statement seems to refer to ref 20, not 28.

Response: That is correct, thank you. We changed this to ref 20.

Response to Reviewer #3

Determining the absolute number of molecules in a protein complex is not easy. Instead, a set of stochastic single-molecule event data reveal the same result with ensemble data. For example, it has been suggested that a certain number of single-molecule events can determine the boundaries of the complexes in superresolution images (Patterson et al. 2010).

The author's main claim is to precisely determine the absolute molecular densities of talin-1, kindlin and integrin in FA using the coupled qSMCL + DNA-PAINT technique, which has not been reported yet in this field. In principle, the authors used the calibrated binding kinetics between DNA origami nanostructures called "docking strands" and "imager strands" labeled by dye or cy3 respectively, which was simultaneously imaged next to the cells expressing talin-halo tag and targeted with same soluble "imager strands" that penetrated the cell membrane. Then, based on Poisson process measurement of tali-1 spatial distribution, the author performed random simulations for clustering to validate the experimentally determined density and spatial distribution length, called NND.

The combined technique applied to this issue is unique because it allows the system to be controlled using previously proven DNA origami engineering. However, it is still unclear to me whether the number of talin-1 and kindlin-2 in FA is absolute or not, at this point. The authors assumed that the binding kinetics between DAN-PAINT docking strands and DNA-imager strands are the same in solution and fixed cell cytoplasm, which may differ in terms of "binding time", which subsequently affects the binding frequency to the same binding site. Viscosity, concentration and physical hinderance should be considered in solution versus the cell membrane that is critically change binding kinetics. One way to validate this problem is for the authors to compare the dwell time (photobleaching time not photo blinking event) of DNA-imager strands in origami and the cell-membrane.

Response: We thank the reviewer for careful evaluating the manuscript and providing very constructive feedback. In response to the reviewer's comments, we have modified the main text and the methods section, which should be easier to follow now. We also included additional Extended Data Figures (now Supplementary Figures) and two Supplementary Tables. Together, we believe that this has significantly strengthened the manuscript.

Detail questions:

Result 1: Determine absolute # of talin-1 in FA.

1) The authors convert the observed time-trace of the binding frequency to the absolute numbers of binding sites. In detail (not described in the manuscript, but referenced to previous papers), the authors used the mean off-time (τd^) of cy3 or the dye tagged to DNA-imager to determine the binding site number. τd^* was derived from the distribution of the waiting-time (fluorescent off time) in the time-trace of the binding frequency. Another parameter needed to calculate the binding site number is the influx rate that is equal to $k_{on} \times [\text{imager concentration}]$. Then finally, the number of binding sites is equal to $(\tau d^* \times k_{on} \times [\text{imager concentration}])^{-1}$. In this equation, the*

author must prove that the k_{on} and [DNA-imager concentration] of solution and cell cytoplasm are the same. In general, the binding kinetics of proteins in cellular systems is complex and affected by not only the concentration of reactants, but also geometric factors and the presence of competitors. In addition, the rate of diffusion of DNA-imager in the fixed cell cytoplasm may differ from that of the medium, since it is another critical factor in protein-protein binding frequency. The authors should discuss this issue.

Response: This is undoubtedly an important issue but please note that the effects of the microenvironment on qPAINT performance have been thoroughly tested previously (Jungmann et al, *Nature Methods*, 2016). Those experiments revealed very similar binding kinetics in different subcellular locations and our experiments are highly consistent with these findings. We compared the number of binding events on single origami binding sites with the number of binding events of manual selected talin localization clouds and observed the same absolute number of visits in both cases (see Figure 1 for reviewer #3). This indicates that the ON-rates are very similar, as expected. Additionally, we compared the ON-time of single DNA origami binding events with those measured in cells and, again, observed similar values (Figure 1c for reviewer #3). To make this point clearer, we include the following data in the Supplementary Information, now Supplementary Fig. 3.

Figure 1 for reviewer #3. Comparison of qPAINT signal in the cytoplasm and on DNA-Origami. a. Representative binding event histories of a single DNA origami binding site (top) and a talin localization cloud (below). **b.** Statistical analysis of binding events on DNA origami structures can be used to calibrate the mean number of binding events per docking site. Subsequent comparison to the event numbers per localization cloud reveals that indeed a talin localization cloud contains one docking strand and thus one talin-1 protein. **c.** Analysis of the ON-time of single binding events on DNA origami and in the cell obtain similar values.

2) How does the author determine influx rate? They used DNA-imager between 250 pM and 2.5 nM. Was k_{on} determined in this system?

Response: The qPAINT measurements were performed with the P3 imager strand using a concentration of 2.5 nM, and k_{ON} was extracted from the visits of the imager strand on the DNA origami. The integrin-talin-kindlin colocalization experiment used the previously described R1 and R2 strands (Strauss and Jungmann, *Nature Methods*, 2020) at a concentration of 250 pM. To ensure that the different experimental conditions are comparable, we quantified their performance in terms of detected binding events and obtained resolution. As shown below, experiments are highly consistent.

Figure 2 for reviewer #3. Effects of different imager strands are negligible. a-c. Analyzing binding events on DNA origami and talin localization clouds using different imager strands reveal highly homogenous distribution of visits per localization cloud in all cases. **d-f.** Fitting of a Gaussian distribution to the center of mass aligned single localization clouds ($n=300$) of different proteins and DNA-strands show comparable high resolution ($\sigma_{ori-P3} = 7\text{nm}$; $\sigma_{tln-P3} = 7.4\text{nm}$; $\sigma_{tln-P1} = 9.6\text{nm}$; $\sigma_{k2-P3} = 7.9\text{nm}$; $\sigma_{k2-R2} = 6.3\text{nm}$; $\sigma_{tln-R1} = 6.6\text{nm}$; $\sigma_{9EG7-P3} = 6.8\text{nm}$). Please note that all average images have the same intensity values leading to a seemingly brighter signal for the 9EG7 average, caused by multiple docking sites per antibody. Data were fit with a Gaussian model. Scale bars: 10 nm.

3) Increasing the observation time increases the total number of binding events due to the irreversible binding/unbinding kinetics between the Halo tagged-Talin and DNA-imager strands. How does the author consider the rebinding events at the previous binding site, which would overcount the absolute number of binding sites?

Response: Please note that rebinding of imager strands to the same site is actually a great advantage of DNA-PAINT microscopy. It allows to discriminate specific from non-specific binding events, which do not show repetitive binding. As discussed below, this feature was used to filter the data and isolate specific signals. As this is an important issue, we tried to clarify this point by generating a new Supplementary Fig. 1, where we explain the principle of DNA-PAINT in more detail. Please note that during a qPAINT experiment the repetitive visits occur at the DNA origami as well as the Halo-tagged talin molecule. This does not lead to overcounting of binding sites.

4) How do the authors discriminate between one or multiple binding sites based on the digitized fluorescence signal? For example, in Figures 1 J and K, both single and triple bonds show the same on/off digitized fluorescence signal. Is the intensity for triple binding 3-time higher than for the single bonded fluorescence signal (it looks like)? Or do the # of events contribute to fluorescence signal intensity (Figure 1 k is 3-time higher than at Figure 1 J (also it looks like)? When authors show the raw data of the intensity profile for the event traces, the audience can easily understand the processing protocol instead of just showing only the digitized fluorescence intensity trace.

Response: Single- and multiple binding sites are distinguished by the number of detected localization events. The fluorescence intensities of the individual signals, as expressed as mean fluorescence per localization, are indistinguishable. To illustrate this important point, we now include a new data set into Figure 1 (Fig. 11). The included image is also shown below (Figure 3 for reviewer #3).

Figure 3 for reviewer #3. Statistical analysis of the mean photon counts per individual localization event reveals highly similar values for single (Fig. 1j) and triple (Fig. 1k) binding sequences.

Result 2: Cluster analysis and theoretical simulation

1) How do the authors eliminate unspecified single molecule events when using DBSCAN to detect clusters? The criteria used by authors should be explained along with their reasoning, since every single-molecule event on the screen is a real event. This issue is important while thresholding molecular events in SMLM images and in sptPALM signal. For example, in a living cell system, a signal for one-frame at a high video rate (10 - 30 fps) can be considered as a non-specific event and filtered out. Indeed, in principle, the portion of single frame events is significant in fluorescent labelled imaging data, and it is difficult to estimate how many of such events are involved in the total data of the fixed sample. Instead, the authors used mean frame and std frame filtering when removing non-specific events, which are tail parts in frame distributions (Extended data figure 2 h-1). Are the distribution of mean frame and std frame the same in solution (i.e., DNA origami control binding) and in cell cytoplasm (talin-halo:DNA-imager)?

Response: We agree that the data analysis procedure could be explained in more detail. Therefore, we modified the main text and included a more extensive paragraph into the methods section (under “Data processing and analysis”). Furthermore, we update the associated Extended Data Figure (now Supplementary Fig. 4) to clarify how the data analysis was conducted. In brief, the algorithm assumes that the specific binding of an imager strand to its complementary target should occur throughout the image acquisition and in a regular fashion. Thus, the mean frame of a typical data set of 80.000 frames should be at around 40.000 frames. To remove non-repetitive binding events, caused by random blinking or unspecific association, the mean frame value of all detected localization clouds was fitted and a cut-off value at +/- the standard deviation was set. To remove unspecific signals that occur, accidentally, at around 40.000 frames, the standard deviation frame filter is used. By plotting the standard deviation of the mean frame values, all data below a cut-off value are disregarded. Please note that this procedure has been described previously by Wade et al, *Nano Letters*, 2019. Figure 4 for reviewer #3 (shown below) illustrates this procedure and it demonstrates that the distribution of mean frame and standard deviation frames are similar for DNA origami and talin-Halo measurements.

Figure 4 for reviewer #3. Distribution of mean frame and standard deviation frames a. Repetitive transient binding to DNA origami and single sites in the cells leads to a mean frame of roughly half the number of total frames in the acquisition window ($mean_{Origami} = 45.380 \pm 18.864$;

$mean_{Cell} = 36.005 \pm 15.394$). **b.** *Non-repetitive binding will result in a mean frame located within the frames of their random appearance distributed in the acquisition window and thus resulting in a small standard deviation.*

2) How did the author perform the segmentation process shown in the extended data Figure 2g-i? Unlike DBSCAN control origami data (Extended data figure 3), there are large clusters, some of which are interconnected (Extended data figure 2 g). It is a good idea to briefly discuss the automatic cluster identification algorithm.

Response: We updated the associated Extended Data Figure (now Supplementary Fig 4) and its figure legend, and also include two additional Supplementary Tables to better explain the cluster detection algorithm. Please note that the interconnection of clusters in Supplementary Fig. 4g are due to slightly larger radius that is initially used to identify a maximal number of initial clusters before filtering. This radius is reduced in subsequent steps to avoid artificial merging of signals.

3) Based on the results of this section, the authors concluded that the spatial distribution of talin-1 in FA is governed by the molecular density instead of the molecular ruler function of talin-1 proposed previously. They calculated the density of talin-1 as 600 molecules/ μm^2 in FA, which “completely” packed in the adhesion (Talin-1 is 270kDa, which is 10x larger than GFP, assuming even a linear structure of 4.8 nm width and 420 nm length). Thus, this excludes the existence of other molecules which is normally un-realistic because molecule-molecule interactions occur in a mixture of different proteins as shown in figures 3 and 4 (overlapping between talin-1, kindlin, and integrin $\beta 1$). Is this artifact caused by a reversible binding event at the same binding site as mentioned above? This issue is related to the following questions.

Response: Previous work (Kanchanawong et al, *Nature*, 2010) has demonstrated that talin associates with its N-terminal FERM domain at the plasma membrane where integrin receptors are bound. The long C-terminal rod-domain, however, is directed towards the cytoplasm at an angle of approximately 15 % (Liu et al, *PNAS*, 2015) to engage the actin cytoskeleton. It therefore seems that only the comparably small talin FERM domain, with dimension of 4 nm x 7 nm x 16 nm (Eliot et al, *Structure*, 2010), needs to be accounted for at the plasma membrane, leaving sufficient space to allow for the localization of other FA proteins.

Result 3: Double labelling imaging of Kindlin-2 and Talin shows significant overlap in the clusters shown in Figure 3. In this case, is the absolute number of each molecule the same as determined in the single-color imaging data shown in Figure 1?

Response: Yes, we compared the binding events from Fig. 1 and Fig. 3 and confirmed, that the absolute numbers of molecules per localization cloud is the same. As shown above in Figure 2d-f for reviewer #3, talin and kindlin histograms show a unimodal distribution, indicating a true molecular resolution. Please also see figure 3 for reviewer #2, which explains the display mode in more detail.

Result 4: Again, in the triple DNA-PAINT imaging of talin, kindlin and integrin $\beta 1$, the mutual overlapped areas between clusters are significant. Can the author further analyze the absolute number of copies of each component in this overlapped area? Since this paper's primary claim relates to this issue (determining absolute number of molecules in FA), this question will be of interesting to the audience. The lateral distribution of these three components is not a big deal as the 3D spatial information is still missing in the 2-D projection images.

Response: As discussed above, the unimodal distribution of binding events demonstrates that single talin and kindlin proteins are detected in Figure 1 (talin), Figure 3 (talin and kindlin) and Figure 4 (integrin, talin, and kindlin). Please note that due to the NHS-based labeling of the antibody, we cannot claim absolute numbers in case of integrin, which is now also clearly stated in the figure legend of Fig. 4. We note, however, that the resolution of the integrin data is highly similar to talin and kindlin data ($\sigma_{K2} = 6.3$ nm, $\sigma_{tln} = 6.6$ nm, $\sigma_{itg} = 6.8$ nm), which strongly indicates that single integrins are detected.

Overall, it is a good idea to consider publishing this paper if the authors can address some of the critical issues raised above. This is because experimentalists generally use relative comparisons rather than focusing on the absolute numbers due to many unresolved factors, especially in protein-protein kinetics in cells (Oh et al. 2012). And in vivo and fixed samples may have very different properties that affect binding kinetics and consequently influence measurements of the absolute copy number.

Oh, D., M. Ogiue-Ikeda, J. A. Jadwin, K. Machida, B. J. Mayer, and J. Yu. 2012. 'Fast rebinding increases dwell time of Src homology 2 (SH2)-containing proteins near the plasma membrane', Proc Natl Acad Sci U S A, 109: 14024-9.

Patterson, G., M. Davidson, S. Manley, and J. Lippincott-Schwartz. 2010. 'Superresolution imaging using single-molecule localization', Annu Rev Phys Chem, 61: 345-67.

Response: Once again, we thank the reviewer for these very constructive comments.

Reviewers' Comments:

Reviewer #2:

Remarks to the Author:

The authors have generally constructively responded to the question raised, many thanks.

A few, mostly minor, points remain, prompted by the responses.

1) re discussion of the applicability of the approach raised and the author response: *"We thank the reviewer for this suggestion and added these details into the discussion. The new segment reads: "The approach should be adaptable to all SMLM techniques obtaining molecular resolution such as MINFLUX3 , Expansion SMLM29 , and – as shown in this study PAINT-based approaches. The experiments require careful calibration to determine the number of molecules per localization cloud, which can be realized by qPAINT measurements using DNA origami or, potentially, recently developed localization-based fluorescence correlation spectroscopy (IbFCS)³⁰ . Furthermore, the theoretical simulations require an estimate of the labelling and detection efficiencies, and biological controls are key to validate the calculated degree of protein-protein association in the given subcellular context."*

Two issues remain:

(a) the term "molecular resolution" is unfortunately often used loosely in the field of super-resolution and often applied to any of the PALM/STORM etc type SMLM methods, regardless of actual resolution achieved. The authors need to stress a clearer term, e.g. "true molecular resolution, i.e. allowing segmentation of point clouds for individual labeled proteins" so that readers fully appreciate the stringency of this requirement.

(b) the quantitative labelling requirement, a "precise number of DS per molecule labeled" is missing from the list of requirements. Please add clearly.

2) The authors respond: *"As now indicated in Supplementary Fig. 1b, picking is a manual selection process. ... however, it has a detection efficiency of 100 %"*

Few, if any, manual procedures have 100% detection efficiency. This statement is therefore a quite strong claim and requires evidence that confirms this fact. This also needs to establish that inter-operator variability is negligible.

3) Several replies mention important additional information but state it only for the reviewers sake. These statements contain important information for replication by others and/or interpretation of the data shown and therefore have to be added to the manuscript for all readers. These are:

a) *"Yes, we used the convex hull to visualize the shape of DBSCAN-detected localization clouds. The overlap of the coloured regions, as seen in Supplementary Fig. 5c (former Extended Data Fig. 3c), depends on the chosen intensity. Please note that this is just a way of visualization (see Fig. 3 for reviewer #2), which does not affect the complex formation or distance analysis calculation."*

The information about displaying convex hulls of point clouds needs to be added to the respective figure legends as information needed to interpret panels shown.

b) *"We agree with the reviewer that MinPts is an important parameter and we included another table into the Supplementary data, where all relevant parameters are shown. ...On average, we observed ~120 localizations per localization cloud and this value was virtually unchanged over three years of imaging (see Figure 2 for reviewer #2)."*

The MinPts values chosen are now in the table (values of 15-20). The information in "Figure 2 for reviewer #2", such as panel a, is relevant for making these choices and should be supplied as supplementary data for all readers motivating these parameter values.

Reviewer #3:

Remarks to the Author:

The authors have addressed many of my concerns and I feel that it is acceptable.

Authors Response for Manuscript NCOMMS-20-33828A

Quantitative single-protein imaging reveals molecular assembly of integrin, talin, and kindlin during cell adhesion

Lisa S. Fischer, Christoph Klingner, Thomas Schlichthaerle, Maximilian T. Strauss, Ralph Böttcher, Reinhard Fässler, Ralf Jungmann and Carsten Grashoff

Response to Reviewer #2

The authors have generally constructively responded to the question raised, many thanks.

A few, mostly minor, points remain, prompted by the responses.

Response:

We thank the reviewer for carefully evaluating our revised manuscript and the constructive suggestions. We have added further information into the main text and the methods section, and we included an additional Supplementary Figure. We believe that this will clarify the remaining issues and hope that the reviewer can now fully support the publication of our study.

1) re discussion of the applicability of the approach raised and the author response: “We thank the reviewer for this suggestion and added these details into the discussion. The new segment reads: “The approach should be adaptable to all SMLM techniques obtaining molecular resolution such as MINFLUX3 , Expansion SMLM29 , and – as shown in this study PAINt-based approaches. The experiments require careful calibration to determine the number of molecules per localization cloud, which can be realized by qPAINt measurements using DNA origami or, potentially, recently developed localization-based fluorescence correlation spectroscopy (IbFCS)30 . Furthermore, the theoretical simulations require an estimate of the labelling and detection efficiencies, and biological controls are key to validate the calculated degree of protein-protein association in the given subcellular context.””

Two issues remain:

(a) the term “molecular resolution” is unfortunately often used loosely in the field of super-resolution and often applied to any of the PALM/STORM etc type SMLM methods, regardless of actual resolution achieved. The authors need to stress a clearer term, e.g. “true molecular resolution, i.e. allowing segmentation of point clouds for individual labeled proteins” so that readers fully appreciate the stringency of this requiremen

(b) the quantitative labelling requirement, a “precise number of DS per molecule labeled” is missing from the list of requirements. Please add clearly.

Response: We fully agree with both suggestions and changed the wording to ‘*true single protein resolution*’ and included a statement on the quantitative labeling requirements. The text now reads:

‘The approach should be adaptable to all SMLM techniques obtaining true single protein resolution such as MINFLUX³, Expansion SMLM²⁹, and – as shown in this study – PAINt-based approaches. The experiments require a defined number of docking strands per target protein to determine the number of molecules per localization cloud using qPAINt calibration or, potentially, recently developed localization-based fluorescence correlation spectroscopy (lbFCS)³⁰.’

2) The authors respond: “As now indicated in Supplementary Fig. 1b, picking is a manual selection process. ... however, it has a detection efficiency of 100 %”

Few, if any, manual procedures have 100% detection efficiency. This statement is therefore a quite strong claim and requires evidence that confirms this fact. This also needs to establish that inter-operator variability is negligible.

Response: We apologize for this overly optimistic statement and agree that manual picking is probably also not entirely flawless. What we wanted to communicate in our response to the reviewer is that the manual selection process serves as the gold standard for signal detection and is therefore used in this study to estimate the DBSCAN detection efficiency. Please note that we had not included the statement on 100 % detection efficiency in the original manuscript. To clarify this point, however, we rephrased the methods section stating that we assumed a 100 % detection efficiency for the manual selected data to estimate the DBSCAN detection efficiency. We believe that this better describes our reasoning and will allow the reader to better understand how the values for DBSCAN detection efficiency were derived.

3) Several replies mention important additional information but state it only for the reviewers sake. These statements contain important information for replication by others and/or interpretation of the data shown and therefore have to be added to the manuscript for all readers. These are:

(a) “Yes, we used the convex hull to visualize the shape of DBSCAN-detected localization clouds. The overlap of the coloured regions, as seen in Supplementary Fig. 5c (former Extended Data Fig. 3c), depends on the chosen intensity. Please note that this is just a way of visualization (see Fig. 3 for reviewer #2), which does not affect the complex formation or distance analysis calculation.”

The information about displaying convex hulls of point clouds needs to be added to the respective figure legends as information needed to interpret panels shown.

Response: We followed the reviewer suggestion and added the information about visualization of our data to the respective figure legends. In addition, we included a small paragraph in the methods section under ‘*Data processing and analysis*’. The end of this paragraph now reads: ‘*The visualization is based on the convex hull of DBSCAN detected, grouped localization clouds. Please note that the overlap of the colored regions depends on chosen intensity, which does not affect the complex formation and distance analysis calculation.*’

(b) “We agree with the reviewer that MinPts is an important parameter and we included another table into the Supplementary data, where all relevant parameters are shown. ...On average, we observed ~120 localizations per localization cloud and this value was virtually unchanged over three years of imaging (see Figure 2 for reviewer #2).”

The MinPts values chosen are now in the table (values of 15-20). The information in “Figure 2 for reviewer #2”, such as panel a, is relevant for making these choices and should be supplied as supplementary data for all readers motivating these parameter values.

Response: We included the requested information as an additional supplementary figure (now Supplementary Fig. 4) and we specifically refer to these data set in the main text, which now reads: ‘*To overcome this challenge, we next developed an automated data processing procedure that was based on a DBSCAN cluster detection²⁴ using constant parameter sets enabled by highly reproducible data acquisition (Supplementary Fig. 4). The algorithm includes two consecutive filtering steps to remove all unspecific signals and calculate the nearest neighbor distance (NND) between individual proteins (Supplementary Fig. 5).*’

Response to Reviewer #3

The authors have addressed many of my concerns and I feel that it is acceptable.

Response: We thank the reviewer once again for evaluating our manuscript and providing very constructive comments.